# Analysis of coupling in geographic information systems based on WASPAS method for bipolar complex fuzzy linguistic Aczel-Alsina power aggregation operators

Zeeshan Ali[1], Khizar Hayat[2]*, Dragan Pamucar [3,4,5]*

1 Department of Information Management, National Yunlin University of Science and Technology, Douliou, Yunlin, Taiwan, 2 Department of Mathematics, University of Kotli, AJ&K, Pakistan, 3 Széchenyi István University, Győr, Hungary, 4 Department of Industrial Engineering & Management, Yuan Ze University, Taoyuan City, Taiwan, 5 Department of Mechanics and Mathematics, Western Caspian University, Baku, Azerbaijan

* khizarhayat@uokajk.edu.pk (KH); dragan.pamucar@fon.bg.ac.rs (DP)

**Data Availability Statement:** All the data are included in the manuscript.

**Funding:** The author(s) received no specific funding for this work.

## Abstract

The model of bipolar complex fuzzy linguistic set is a very famous and dominant principle to cope with vague and uncertain information. The bipolar complex fuzzy linguistic set contained the positive membership function, negative membership function, and linguistic variable, where the technique of fuzzy sets to bipolar fuzzy sets are the special cases of the bipolar complex fuzzy linguistic set. In this manuscript, we describe the model of Aczel-Alsina operational laws for bipolar complex fuzzy linguistic values based on Aczel-Alsina t-norm and Aczel-Alsina t-conorm. Additionally, we compute the Aczel-Alsina power aggregation operators based on bipolar complex fuzzy linguistic data, called bipolar complex fuzzy linguistic Aczel-Alsina power averaging operator, bipolar complex fuzzy linguistic Aczel-Alsina power weighted averaging operator, bipolar complex fuzzy linguistic Aczel-Alsina power geometric operator, and bipolar complex fuzzy linguistic Aczel-Alsina power weighted geometric operator with some dominant and fundamental laws such as idempotency, monotonicity, and boundedness. Moreover, we initiate the model of the Weighted Aggregates Sum Product Assessment technique with the help of consequent theory. In the context of geographic information systems and spatial information systems, coupling aims to find out the relationships among different components within a geographic information system, where coupling can occur at many stages, for instance, spatial coupling, data coupling, and functional coupling. To evaluate the above dilemma, we perform the model of multi-attribute decision-making for invented operators to compute the best technique for addressing geographic information systems. In the last, we deliberate some numerical examples for comparing the ranking results of proposed and prevailing techniques.

**Competing interests:** The authors declare that they have no known competing financial interests or personal relationships that could have appeared to influence the work reported in this paper.

# 1. Introduction

The type and degree of coupling in a geographic information system [1] can affect its performance, such as scalability and flexibility. In the meaning of geographic information systems and spatial data systems, coupling main theme is to evaluate the interrelationships between different components within a geographic information system [1]. Further, finding the best optimal based on some attributes among the collection of alternatives is a very complex task, because many people have evaluated such kind of procedure based on classical set theory, where the decision-making procedure, MADM procedure is very famous for depicting vague and unreliable information, but they lost a lot of information due to limited options [2]. To handle such kinds of problems, Zadeh [3] exposed the fuzzy set (FS). FSs have a truth grade defined from universal set to unit interval, such as $\Xi_{\varpi}(\mathbb{X}) \in [0, 1]$. Furthermore, FS theory has only a truth grade, supporting grade, or positive grade, but in many situations, we noticed that the truth grade is not enough for depicting vague and complex information, because in many cases we noticed that the involvement of negative, support against, and negative information. To handle such kind of problems, Zhang [4] introduced the bipolar FS (BFS) theory. BFSs is the modified version of the FSs theory, because FSs contained just a simple truth grade, but the BFSs theory contained the positive truth grade "$\Xi_{\varpi}(\mathbb{X}) \in [0, 1]$" and negative truth grade "$\Psi_{\varpi}(\mathbb{X}) \in [0, 1]$, where the idea of FS is the dominant part of BFSs. The involvement of two-dimensional theory has played an important role during decision-making procedures and genuine-life problems, for instance, to buy a car from any company, we will put our opinion in the following shape, name of the car and production data of the car, for managing such kind of problems, the truth grade of FSs is not enough because they just deal with one-dimensional information. For this, Ramot et al. [5] exposed the complex FS (CFS), where the truth grade in CFS is computed in the shape: $\Xi_{\varpi}(\mathbb{X}) + i\Omega_{\varpi}(\mathbb{X})$, where $\Xi_{\varpi}, \Omega_{\varpi} : \mathbb{X} \to [0, 1]$. Further, Mahmood and Rehman [6] exposed the novel technique of bipolar CFSs (BCFSs), where the positive membership grade and negative membership grade are as follows: $\Xi_{\varpi}(\mathbb{X}) + i\Omega_{\varpi}(\mathbb{X})$ and $\Psi_{\varpi}(\mathbb{X}) + i\Phi_{\varpi}(\mathbb{X})$ with a characteristic, such as $\Xi_{\varpi}(\mathbb{X}), \Omega_{\varpi}(\mathbb{X}) \in [0, 1]$ and $\Psi_{\varpi}(\mathbb{X}), \Phi_{\varpi}(\mathbb{X}) \in [-1, 0], i = \sqrt{-1}$, these two grades are the major parts of the BCFSs. BCFSs are very well-known due to their unlimited features and because of their structure, the FSs, CFSs, and BFSs are the special parts of the BCFSs. The linguistic set theory was initiated by Zadeh [7–9], where linguistic variables are used in many situations of life, for instance, if we talked about the weather, we used the following information, such as very cold, cold, normal, hot, very hot are given linguistic terms.

## 1.1. Literature review

The decision-making technique is a very reliable and dominant technique for addressing the best optimal among the collection of finite information. Because of unlimited ambiguity and uncertainty, the decision-making technique has not worked dominantly because of a crisp set. To address the above problems FS has been proposed. FS theory has a lot of applications in many fields, for instance, fuzzy n-soft sets were invented by Akram et al. [10], fuzzy superior Mandelbrot sets were derived by Mahmood and Ali [11], and hesitant fuzzy n-soft sets were derived by Akram et al. [12]. Further, the model of FS has only a membership function, but in various genuine-life situations, we need the technique of BFS, because it covers the membership function in the form of positive and negative ways. After the utilization of the BFSs, many well-known and dominant ideas have been proposed by different scholars, for instance, bipolar fuzzy metric space was invented by Zararsız and Riaz [13], and analysis of ideal in BCI-algebra in BFSs was discovered by Abughazalah et al. [14], and bipolar vague soft sets were presented by Sakr et al. [15]. Moreover, CFSs are more superior and effective than FSs, because

of their features, and due to this reason, many people have used them in many areas, for instance, distance measures for CFSs were proposed by Liu et al. [16], complex dual type-2 hesitant fuzzy sets were proposed by Mahmood et al. [17], and complex multi-fuzzy hypersoft sets was initiated by Saeed et al. [18]. After the utilization of the BCFSs, many well-known and dominant ideas have been proposed by different scholars, for instance, bipolar complex fuzzy soft sets were derived by Gwak et al. [19] and bipolar complex linear systems were presented by Akram et al. [20]. Furthermore, linguistic sets have been utilized by different scholars in different fields, for instance, fuzzy linguistic sets (FLS) [21], complex fuzzy linguistic sets (CFLS) [22, 23], bipolar fuzzy linguistic sets (BFLSs) [24], and bipolar complex fuzzy linguistic sets (BCFLSs) [25]. Further, Yager [26] evaluated power operators for crisp values. Moreover, power-geometric operators for classical set theory were initiated by Xu and Yager [27]. Zavadskas et al. [28] exposed the WASPAS method for crisp values. Mardani et al. [29] presented the WASPAS theory for FSs and Jaleel [30] exposed it for BCFSs. Bi et al. [31] initiated the arithmetic operators for CFSs and Hu et al. [32] derived the power operators for CFSs. Jana et al. [33] exposed the Dombi operators for BFSs. Mahmood et al. [34] evaluated the aggregation operators for BCFSs. Further, Aczel-Alsina t-norm (AATN) and Aczel-Alsina t-conorm (AATCN) were proposed by Aczel and Alsina [35]. Moreover, Mahmood et al. [36] presented the Aczel-Alsina operators for BCFSs. Further, some fundamental techniques and methods are described in the following, for instance, Schweizer-Sklar operators [37], the VIKOR technique [38], the SWARA-COPRAS technique [39], renewable energy resources [40], a best-worst technique [41], decision-making technique [42], DEMATEL-ISM integration method [43], IFSs and their applications [44], BHARAT decision-making model [45], Hamy mean operators [46], Aczel-Alsina operators [47], analysis of decision accuracy in DEMATEL technique [48], Einstein operators [49], MADM technique based on vague sets [50], Bonferroni mean operators [51], decision-making strategy [52, 53], fuzzy soft code [54], Fermatean fuzzy sets [55], analysis of convexity based on hyper-soft sets [56], Pythagorean fuzzy linear programming [57], analysis of hybrid model based on IFS [58], the parsimonious spherical fuzzy sets [59, 60], the AHP model [61], linear programming [62] and the aggregation operators [63].

## 1.2. Research gap and major problems

The model of FSs to BCFL set is a very common technique that is used for addressing different kinds of problems in genuine-life dilemmas. Further, various kind of operators, methods, and measures was proposed based on it by different scholars. Additionally, during the analysis and revision of the existing techniques, we observed that every decision-maker has problems with the help of major queries, for instance.

1. Problem 1: How do we define new operational laws?

2. Problem 2: How do we aggregate the collection of the finite number of alternatives into a singleton set?

3. Problem 3: How do we get the excellent optimal among the collection of alternatives?

In the consideration or availability of the above problems, no one can derive accurate results because of ambiguity and limitations. Anyhow, the model of the WASPAS technique and Aczel-Alsina power operators for bipolar complex fuzzy linguistic sets are the best solutions to the above problems. The model of the BCFL set has been proposed, but no one can define any kind of operators or any type of method based on it because the structure of the BCFL set is very complex due to linguistic terms, where the positive and negative membership

function is also computed in the shape of complex-number, so it is a quite complex and challenging task for scholars to define any kind of operators or method based on it. The major reason for the construction of the WASPAS method is that with the help of the WASPAS technique, we can easily evaluate the best optimal among the collection of information. But there are still problems, if we have a collection of a finite number of values, then what happens? For evaluating such kind of dilemmas, we propose the technique of Aczel-Alsina power operators, because with the help of power operators, we evaluate the weight vectors, if we use the unknown weight vectors then we may get the wrong result because of ambiguity and complications, but if we have known weight vectors, then we will get accurate results, therefore, by using the Aczel-Alsina power operator, we can easily aggregate the collection of information into a singleton set, which can help in the implementation of the WASPAS model to address the problems of MADM technique.

## 1.3. Motivation/Advantages/Major contributions of the proposed methods

Fuzzy set theory contains a very wide range of applications in many fields and various scholars have developed different kinds of extensions based on FS theory, where BCFL is one of them. The model of the BCFL set is a very reliable and flexible model for coping with uncertain and vague information, because of complex-valued positive and complex-valued negative membership functions with linguistic term sets. Further, the model of FS, linguistic term set, BFS, CFS, and BCFS are the sub-part of the BCFL sets. Moreover, the technique of Aczel-Alsina aggregation operators is also the modified version of many existing operators, called maximum aggregation operators, minimum aggregation operators, Drastic aggregation operators, and algebraic aggregation operators. Additionally, the power aggregation operators are also a dominant technique for aggregating the collection of information into a singleton set. Motivated by the structure and inspired form their advantages, the major advantages of the presented techniques are listed below:

1. Maximum, minimum, Drastic, Algebraic, power, Aczel-Alsina aggregation operators, and WASPAS method for fuzzy information.

2. Maximum, minimum, Drastic, Algebraic, power, Aczel-Alsina aggregation operators, and WASPAS method for bipolar fuzzy information.

3. Maximum, minimum, Drastic, Algebraic, power, Aczel-Alsina aggregation operators, and WASPAS method for linguistic term information.

4. Maximum, minimum, Drastic, Algebraic, power, Aczel-Alsina aggregation operators, and WASPAS method for complex fuzzy information.

5. Maximum, minimum, Drastic, Algebraic, power, Aczel-Alsina aggregation operators, and WASPAS method for bipolar fuzzy information.

6. Maximum, minimum, Drastic, Algebraic, power, Aczel-Alsina aggregation operators, and WASPAS method for bipolar complex fuzzy information.

7. Maximum, minimum, Drastic, Algebraic, power, Aczel-Alsina aggregation operators, and WASPAS method for fuzzy linguistic term information.

8. Maximum, minimum, Drastic, Algebraic, power, Aczel-Alsina aggregation operators, and WASPAS method for bipolar fuzzy linguistic term information.

9. Maximum, minimum, Drastic, Algebraic, power, Aczel-Alsina aggregation operators, and WASPAS method for complex fuzzy linguistic term information.

10. Maximum, minimum, Drastic, Algebraic, power, Aczel-Alsina aggregation operators, and WASPAS method for bipolar complex fuzzy linguistic term information.

The above information is the special cases of the proposed techniques. It is clear that the proposed techniques and operators are very reliable and superior because of their features, based on the above advantages, the major contribution of this manuscript is listed below:

1. For evaluating problem 1, we aim to evaluate the novel model of Aczel-Alsina operational laws for BCFL variables.

2. To address problem 2, we aim to initiate the model of the BCFLAAPOA operator, BCFLAAPOWA operator, BCFLAAPOG operator, and BCFLAAPOWG operator.

3. For the simplification of the above operators, we aim to derive the major and fundamental properties of the proposed theory, called idempotency, monotonicity, and boundedness.

4. Using the above operators, which are used for aggregating the collection of information into a singleton set, we compute the WASPAS method based on the initiated operators.

5. For the justification of the above information, we aim to demonstrate the procedure of the MADM technique based on initiated operators for computing the best technique for addressing geographic information systems.

6. Finally, we aim to compare the ranking values of initiated techniques with the ranking values of the existing techniques based on illustrated examples to enhance the worth of the proposed theory. The geometrical interpretation of the proposed theory is discussed in Fig 1.

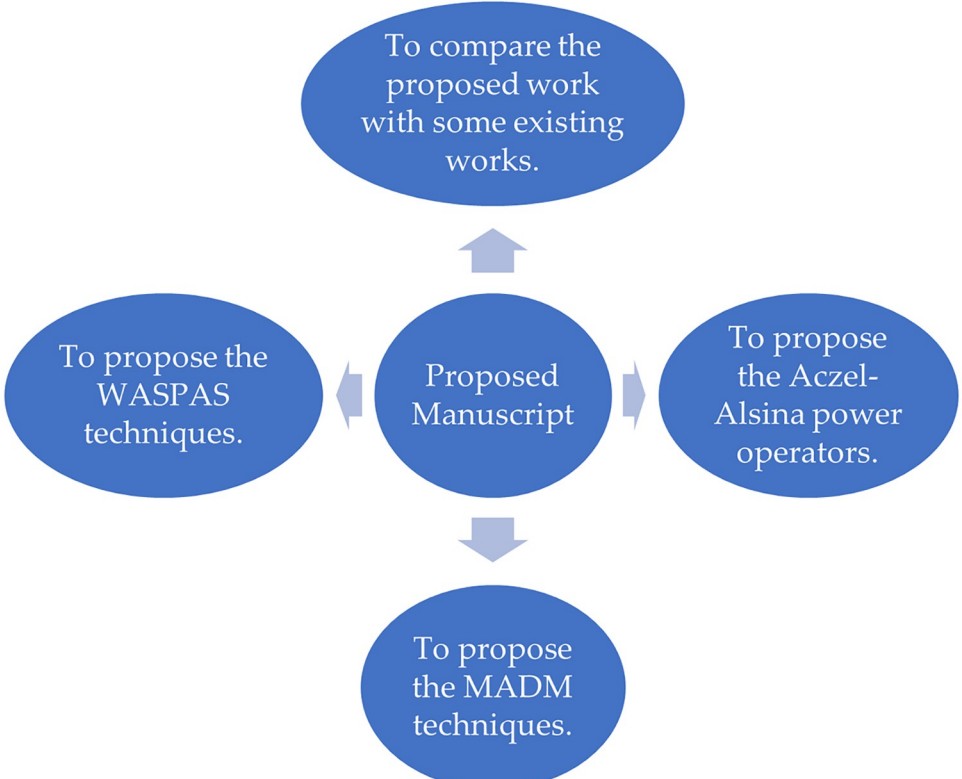

**Fig 1. Geometrical abstract of the proposed theory.**

### 1.4. Summary of the proposed manuscript

This manuscript is computed based on BCFL information with some operators, the shape of this manuscript is arranged in the following form:

1. In Section 2, we discussed the idea of AATN and AATCN. Further, we discussed the idea of a PO averaging (POA) operator and a PO geometric (POG) operator. Moreover, we described the BCFLSs and their basic laws.

2. In Section 3, we presented the Aczel-Alsina operational laws based on BCFL variables. Further, we evaluated the BCFLAAPOA operator, BCFLAAPOWA operator, BCFLAAPOG operator, and BCFLAAPOWG operator. Some fundamental properties are also presented for the above operators.

3. In Section 4, we computed the WASPAS method by using the initiated operators.

4. In Section 5, we demonstrated the procedure of the MADM technique based on initiated operators for computing the best technique for addressing geographic information systems.

5. In Section 6, we compared the ranking values of initiated techniques with the ranking values of the existing techniques based on illustrated examples to enhance the worth of the proposed theory.

6. Some concluding remarks are stated in Section 7.

## 2. Preliminaries

In this section, we revised the technique of AATN, AATCN, POA operator, POG operator, and BCFL set (BCFLS) with some major properties.

**Definition 1:** [35] Consider $\mu_1, \mu_2 \in [0,1]$, thus

$$\mathbb{A}^{\pi}(\mu_1, \mu_2) = \begin{cases} \mathbb{A}^d(\mu_1, \mu_2) & \text{if } \pi = 0 \\ \min(\mu_1, \mu_2) & \text{if } \pi = \infty \\ \mathbb{e}^{-((-\log(\mu_1))^{\pi} + (-\log(\mu_2))^{\pi})^{\frac{1}{\pi}}} & \text{otherwise} \end{cases}$$

$$\mathbb{A}^{@\pi}(\mu_1, \mu_2) = \begin{cases} \mathbb{A}^{@d}(\mu_1, \mu_2) & \text{if } \pi = 0 \\ \max(\mu_1, \mu_2) & \text{if } \pi = \infty \\ 1 - \mathbb{e}^{-((-\log(1-\mu_1))^{\pi} + (-\log(1-\mu_2))^{\pi})^{\frac{1}{\pi}}} & \text{otherwise} \end{cases}$$

Called AATN and AATCN, where $\pi \in [0, \infty]$. Further, the model of drastic t-norm $\mathbb{A}^d(\mu_1, \mu_2)$ and drastic t-conorm $\mathbb{A}^{@d}(\mu_1, \mu_2)$ are described below:

$$\mathbb{A}^d(\mu_1, \mu_2) = \begin{cases} \mu_1 & \text{if } \mu_2 = 1 \\ \mu_2 & \text{if } \mu_1 = 1 \\ 0 & \text{otherwise} \end{cases}$$

$$\mathbb{A}^{@d}(\mu_1, \mu_2) = \begin{cases} \mu_1 & \text{if } \mu_2 = 0 \\ \mu_2 & \text{if } \mu_1 = 0 \\ 1 & \text{otherwise} \end{cases}$$

Where $\mathbb{A}^{\pi}(\mu_1, \mu_2) = \mu_1 * \mu_2$ and $\mathbb{A}^{@\pi}(\mu_1, \mu_2) = \mu_1 + \mu_2 - \mu_1 * \mu_2$ are called algebraic t-norm and algebraic t-conorm.

**Definition 2:** [26, 27] Consider any finite family of non-negative integers $\varpi_1, \varpi_2, \ldots, \varpi_n$, thus

$$POA(\varpi_1, \varpi_2, \ldots, \varpi_n) = \sum_{j=1}^{n} \frac{(1 + \omega(\varpi_j))}{\sum_{j=1}^{n}(1 + \omega(\varpi_j))} \varpi_j$$

$$POG(\varpi_1, \varpi_2, \ldots, \varpi_n) = \prod_{j=1}^{n} (\varpi_j)^{\frac{(1+\omega(\varpi_j))}{\sum_{j=1}^{n}(1+\omega(\varpi_j))}}$$

Called POA operator and POG operator, where the term $\omega(\varpi_j) = \sum_{j \neq k=1}^{n} SP(\varpi_j, \varpi_k)$ and $SP(\varpi_j, \varpi_k) = 1 - DS(\varpi_j, \varpi_k)$, with some conditions:

1. $SP(\varpi_j, \varpi_k) \in [0, 1]$.

2. $SP(\varpi_j, \varpi_k) = SP(\varpi_k, \varpi_j)$.

3. If $SP(\varpi_j, \varpi_k) < SP(\varpi_l, \varpi_m)$ then $DS(\varpi_j, \varpi_k) \geq DS(\varpi_l, \varpi_m)$.

Where,

$$DS(\varpi_j, \varpi_k) = |\varpi_j - \varpi_k|$$

**Definition 3:** [25] Let $\mathbb{X}$ be a universal set. The BCFLS $\varpi^{BC}$ based on $\mathbb{X}$ is illustrated below:

$$\varpi^{BC} = \{(\mathcal{L}_{\mathfrak{l}((\mathbb{X}))}, \Xi_{\varpi}(\mathbb{X}) + i\Omega_{\varpi}(\mathbb{X}), \Psi_{\varpi}(\mathbb{X}) + i\Phi_{\varpi}(\mathbb{X})) : \mathbb{X} \in \mathbb{X}\}$$

Where the positive membership grade and negative membership grade are as follows: $\Xi_{\varpi}(\mathbb{X}) + i\Omega_{\varpi}(\mathbb{X})$ and $\Psi_{\varpi}(\mathbb{X}) + i\Phi_{\varpi}(\mathbb{X})$ with a characteristic, such as $\Xi_{\varpi}(\mathbb{X}), \Omega_{\varpi}(\mathbb{X}) \in [0, 1]$ and $\Psi_{\varpi}(\mathbb{X}), \Phi_{\varpi}(\mathbb{X}) \in [-1, 0], i = \sqrt{-1}$. Further, the representation of linguistic variables is as follows: $\mathcal{L}_{\mathfrak{l}((\mathbb{X}))}$, where $S = \{\mathcal{L}_{\mathfrak{l}((\mathbb{X}))} : \mathfrak{l} = 1, , 2, \ldots, \partial\}$. Finally, we illustrated the simple shape of BCFLN, such as $\varpi_j^{BC} = \varpi_j = (\mathcal{L}_{\mathfrak{l}_j}, \Xi_{\varpi_j} + i\Omega_{\varpi_j}, \Psi_{\varpi_j} + i\Phi_{\varpi_j}), j = 1, 2, \ldots, n$. Moreover, we discussed some operational laws for any two BCFLNs, such as

$$\varpi_1^{BC} \oplus \varpi_2^{BC} = \begin{pmatrix} \mathcal{L}_{\partial\left(\frac{\mathfrak{l}_1}{\partial} + \frac{\mathfrak{l}_2}{\partial} - \frac{\mathfrak{l}_1}{\partial}\frac{\mathfrak{l}_2}{\partial}\right)}, \left(\Xi_{\varpi_1} + \Xi_{\varpi_2} - \Xi_{\varpi_1}\Xi_{\varpi_2}\right) + i\left(\Omega_{\varpi_1} + \Omega_{\varpi_2} - \Omega_{\varpi_1}\Omega_{\varpi_2}\right), \\ -(\Psi_{\varpi_1}\Psi_{\varpi_2}) + i(-(\Phi_{\varpi_1}\Phi_{\varpi_2})) \end{pmatrix}$$

$$\varpi_1^{BC} \otimes \varpi_2^{BC} = \begin{pmatrix} \mathcal{L}_{\partial\left(\frac{\mathfrak{l}_1}{\partial}\frac{\mathfrak{l}_2}{\partial}\right)}, \left(\Xi_{\varpi_1}\Xi_{\varpi_2}\right) + i\left(\Omega_{\varpi_1}\Omega_{\varpi_2}\right), \\ (\Psi_{\varpi_1} + \Psi_{\varpi_2} + \Psi_{\varpi_1}\Psi_{\varpi_2}) + i(\Phi_{\varpi_1} + \Phi_{\varpi_2} + \Phi_{\varpi_1}\Phi_{\varpi_2}) \end{pmatrix}$$

$$\rho\varpi_1^{BC} = \left(\mathcal{L}_{\partial\left(1-\left(1-\frac{\mathfrak{l}_1}{\partial}\right)^{\rho}\right)}, 1 - (1 - \Xi_{\varpi_1})^{\rho} + i(1 - (1 - \Omega_{\varpi_1})^{\rho}), -|\Psi_{\varpi_1}|^{\rho} + i(-|\Phi_{\varpi_1}|^{\rho})\right)$$

$$(\varpi_1^{BC})^{\rho} = \left(\mathcal{L}_{\partial\left(\left(\frac{\mathfrak{l}_1}{\partial}\right)^{\rho}\right)}, (\Xi_{\varpi_1})^{\rho} + i((\Omega_{\varpi_1})^{\rho}), -1 + (1 + \Psi_{\varpi_1})^{\rho} + i(-1 + (1 + \Phi_{\varpi_1})^{\rho})\right)$$

Moreover, we justify the above information with the help of some examples. For this, we consider two BCFL numbers, such as $\varpi_1 = (\mathcal{L}_1, 0.2 + i(0.5), -0.2 + i(-0.4))$ and $\varpi_2 = (\mathcal{L}_2, 0.5 + i(0.7), -0.7 + i(-0.4))$, where the order of linguistic sets is $\partial = 6$ with parameter $\rho = 2$, then

$$\varpi_1^{BC} \oplus \varpi_2^{BC} = \begin{pmatrix} \mathcal{L}_{6\left(\frac{1}{6} + \frac{2}{6} - \frac{1}{6} * \frac{2}{6}\right)}, (0.2 + 0.5 - 0.2*0.5) + i(0.5 + 0.7 - 0.5*0.7), \\ -((-0.2)*(-0.7)) + i(-((-0.4)*(-0.4))) \end{pmatrix}$$

$$= (\mathcal{L}_{2.666}, 0.6 + i(0.85), -0.14 + i(-0.16))$$

$$\varpi_1^{BC} \otimes \varpi_2^{BC} = \begin{pmatrix} \mathcal{L}_{6\left(\frac{1}{6} * \frac{2}{6}\right)}, (0.2*0.5) + i(0.5*0.7), \\ (-0.2 - 0.7 + (-0.2* - 0.7)) + i(-0.4 - 0.4 + (-0.4* - 0.4)) \end{pmatrix}$$

$$= (\mathcal{L}_{0.333}, 0.1 + i(0.35), -0.76 + i(-0.64))$$

$$2*\varpi_1^{BC} = \left( \mathcal{L}_{6\left(1 - \left(1 - \frac{1}{6}\right)^2\right)}, 1 - (1 - 0.2)^2 + i(1 - (1 - 0.5)^2), -|-0.2|^2 + i(-|-0.4|^2) \right)$$

$$= (\mathcal{L}_{1.833}, 0.36 + i(0.75), -0.04 + i(-0.16))$$

$$(\varpi_1^{BC})^2 = \left( \mathcal{L}_{6\left(\left(\frac{1}{6}\right)^2\right)}, (0.2)^2 + i((0.5)^2), -1 + (1 - 0.2)^2 + i(-1 + (1 - 0.4)^2) \right)$$

$$= (\mathcal{L}_{0.1666}, 0.04 + i(0.25), -0.36 + i(-0.64))$$

Further, for any BCFL number, the model of score value and accuracy value is described in the following form, such as

$$SC\left(\varpi_j^{BC}\right) = \frac{\mathfrak{l}_j}{\partial} * \left(\Xi_{\varpi_j} + \Omega_{\varpi_j} + \Psi_{\varpi_j} + \Phi_{\varpi_j}\right) \in [-1, 1]$$

$$AC\left(\varpi_j^{BC}\right) = \frac{\mathfrak{l}_j}{\partial} * \left(\Xi_{\varpi_j} + \Omega_{\varpi_j} - \Psi_{\varpi_j} - \Phi_{\varpi_j}\right) \in [0, 1]$$

With the following conditions, such as If $SC(\varpi_1^{BC}) > SC(\varpi_2^{BC}) \Rightarrow \varpi_1^{BC} > \varpi_2^{BC}$, if $SC(\varpi_1^{BC}) < SC(\varpi_2^{BC}) \Rightarrow \varpi_1^{BC} < \varpi_2^{BC}$, if $SC(\varpi_1^{BC}) = SC(\varpi_2^{BC}) \Rightarrow$, then $AC(\varpi_1^{BC}) > AC(\varpi_2^{BC}) \Rightarrow \varpi_1^{BC} > \varpi_2^{BC}$, if $AC(\varpi_1^{BC}) < AC(\varpi_2^{BC}) \Rightarrow \varpi_1^{BC} < \varpi_2^{BC}$.

## 3. Aczel-Alsina power aggregation operators for BCFLNs

In this section, we describe the model of the BCFLAAPOA operator, BCFLAAPOWA operator, BCFLAAPOG operator, and BCFLAAPOWG operator. Some fundamental properties are also presented for the above operators. For the evaluation of the above operators, we first initiate the model of Aczel-Alsina operational laws based on BCFL information.

**Definition 4:** For any two BCFLNs $\varpi_j^{BC} = (\mathcal{L}_{\mathfrak{l}_j}, \Xi_{\varpi_j} + i\Omega_{\varpi_j}, \Psi_{\varpi_j} + i\Phi_{\varpi_j}), j = 1, 2, \ldots, n$, we describe and define the model of Aczel-Alsina operational laws, such as

$$\varpi_1^{BC} \oplus \varpi_2^{BC} = \left( \begin{array}{c} \mathcal{L}_{\partial\left( 1 - \mathbb{e}^{-\left(\left(-\log\left(1-\frac{\mathfrak{l}_1}{\partial}\right)\right)^\pi + \left(-\log\left(1-\frac{\mathfrak{l}_2}{\partial}\right)\right)^\pi\right)^{\frac{1}{\pi}}} \right)}, \\ \left( 1 - \mathbb{e}^{-((-\log(1-\Xi_{\varpi_1}))^\pi + (-\log(1-\Xi_{\varpi_2}))^\pi)^{\frac{1}{\pi}}} \right) + i\left( 1 - \mathbb{e}^{-((-\log(1-\Omega_{\varpi_1}))^\pi + (-\log(1-\Omega_{\varpi_2}))^\pi)^{\frac{1}{\pi}}} \right), \\ -\left( \mathbb{e}^{-((-\log(|\Psi_{\varpi_1}|))^\pi + (-\log(|\Psi_{\varpi_2}|))^\pi)^{\frac{1}{\pi}}} \right) + i\left( -\left( \mathbb{e}^{-((-\log(|\Phi_{\varpi_1}|))^\pi + (-\log(|\Phi_{\varpi_2}|))^\pi)^{\frac{1}{\pi}}} \right) \right) \end{array} \right)$$

$$\varpi_1^{BC} \otimes \varpi_2^{BC} = \left( \begin{array}{c} \mathcal{L}_{\partial\left( \mathbb{e}^{-\left(\left(-\log\left(\frac{\mathfrak{l}_1}{\partial}\right)\right)^\pi + \left(-\log\left(\frac{\mathfrak{l}_2}{\partial}\right)\right)^\pi\right)^{\frac{1}{\pi}}} \right)}, \\ \left( \mathbb{e}^{-((-\log(\Xi_{\varpi_1}))^\pi + (-\log(\Xi_{\varpi_2}))^\pi)^{\frac{1}{\pi}}} \right) + i\left( \mathbb{e}^{-((-\log(\Omega_{\varpi_1}))^\pi + (-\log(\Omega_{\varpi_2}))^\pi)^{\frac{1}{\pi}}} \right), \\ -1 + \left( \mathbb{e}^{-((-\log(1+\Psi_{\varpi_1}))^\pi + (-\log(1+\Psi_{\varpi_2}))^\pi)^{\frac{1}{\pi}}} \right) + i\left( -1 + \left( \mathbb{e}^{-((-\log(1+\Phi_{\varpi_1}))^\pi + (-\log(1+\Phi_{\varpi_2}))^\pi)^{\frac{1}{\pi}}} \right) \right) \end{array} \right)$$

$$\rho\varpi_1^{BC} = \left( \begin{array}{c} \mathcal{L}_{\partial\left( 1 - \mathbb{e}^{-\left(\rho\left(-\log\left(1-\frac{\mathfrak{l}_1}{\partial}\right)\right)^\pi\right)^{\frac{1}{\pi}}} \right)}, \\ \left( 1 - \mathbb{e}^{-(\rho(-\log(1-\Xi_{\varpi_1}))^\pi)^{\frac{1}{\pi}}} \right) + i\left( 1 - \mathbb{e}^{-(\rho(-\log(1-\Omega_{\varpi_1}))^\pi)^{\frac{1}{\pi}}} \right), \\ -\left( \mathbb{e}^{-(\rho(-\log(|\Psi_{\varpi_1}|))^\pi)^{\frac{1}{\pi}}} \right) + i\left( -\left( \mathbb{e}^{-(\rho(-\log(|\Phi_{\varpi_1}|))^\pi)^{\frac{1}{\pi}}} \right) \right) \end{array} \right)$$

$$(\varpi_1^{BC})^\rho = \begin{pmatrix} \mathcal{L} \left( \partial \left( \mathbb{e}^{-\left(\rho\left(-\log\left(\frac{\mathfrak{l}_1}{\partial}\right)\right)^\pi\right)^{\frac{1}{\pi}}} \right) \right)^{'}, \\ \left( \mathbb{e}^{-(\rho(-\log(\Xi_{\varpi_1}))^\pi)^{\frac{1}{\pi}}} \right) + i \left( \mathbb{e}^{-(\rho(-\log(\Omega_{\varpi_1}))^\pi)^{\frac{1}{\pi}}} \right), \\ -1 + \left( \mathbb{e}^{-(\rho(-\log(1+\Psi_{\varpi_1}))^\pi)^{\frac{1}{\pi}}} \right) + i \left( -1 + \left( \mathbb{e}^{-(\rho(-\log(1+\Phi_{\varpi_1}))^\pi)^{\frac{1}{\pi}}} \right) \right) \end{pmatrix}$$

Using the above initiated operational laws, we aim to construct the technique of Aczel-Alsina power aggregation operators for BCFL values.

**Definition 5:** For any finite family of BCFLNs $\varpi_j^{BC} = (\mathcal{L}_{\mathfrak{l}_j}, \Xi_{\varpi_j} + i\Omega_{\varpi_j}, \Psi_{\varpi_j} + i\Phi_{\varpi_j}), j = 1, 2, \ldots, n$, the model of the BCFLAAPOA operator is described and illustrated below:

$$BCFLAAPOA : Z^n \to Z$$

by

$$BCFLAAPOA\left(\varpi_1^{BC}, \varpi_2^{BC}, \ldots, \varpi_n^{BC}\right) = BCFLAAPOA(\varpi_1, \varpi_2, \ldots, \varpi_n)$$
$$= \frac{(1 + \omega(\varpi_1))}{\sum_{j=1}^n (1 + \omega(\varpi_j))} \varpi_1^{BC} \oplus \frac{(1 + \omega(\varpi_2))}{\sum_{j=1}^n (1 + \omega(\varpi_j))} \varpi_2^{BC} \oplus \ldots \oplus \frac{(1 + \omega(\varpi_n))}{\sum_{j=1}^n (1 + \omega(\varpi_j))} \varpi_n^{BC}$$
$$= \oplus_{j=1}^n \frac{(1 + \omega(\varpi_j))}{\sum_{j=1}^n (1 + \omega(\varpi_j))} \varpi_j^{BC}$$

Where $\omega(\varpi_j) = \sum_{j\neq k=1}^n SP(\varpi_j, \varpi_k)$ and $SP(\varpi_j, \varpi_k) = 1 - DS(\varpi_j, \varpi_k)$, with some conditions:

1. $SP(\varpi_j, \varpi_k) \in [0, 1]$.

2. $SP(\varpi_j, \varpi_k) = SP(\varpi_k, \varpi_j)$.

3. If $SP(\varpi_j, \varpi_k) < SP(\varpi_l, \varpi_m)$ then $DS(\varpi_j, \varpi_k) \geq DS(\varpi_l, \varpi_m)$.

Further,

$$DS\left(\varpi_j, \varpi_k\right) = \frac{1}{2} \left( \frac{|\mathfrak{l}_j - \mathfrak{l}_k|}{\partial} + \frac{1}{4} \left( |\Xi_{\varpi_j} - \Xi_{\varpi_k}| + |\Omega_{\varpi_j} - \Omega_{\varpi_k}| + |\Psi_{\varpi_j} - \Psi_{\varpi_k}| + |\Phi_{\varpi_j} - \Phi_{\varpi_k}| \right) \right)$$

Where $Z^n$ contained the collection of BCFLNs. Further, by using the information in Def. (4), we aim to calculate the aggregated values of the information in Def. (5).

**Theorem 1:** Let $\varpi_j^{BC} = (\mathcal{L}_{\mathfrak{l}_j}, \Xi_{\varpi_j} + i\Omega_{\varpi_j}, \Psi_{\varpi_j} + i\Phi_{\varpi_j}), j = 1, 2, \ldots, n$ be the collection of BCFLNs. Then, by using the information in Def. (4) and Def. (5), we proved that their

aggregated value is also a BCFLN, such as

$$BCFLAAPOA(\varpi_1, \varpi_2, \ldots, \varpi_n)$$

$$= \left( \begin{array}{c} \partial^{1-\mathbb{e}^{-\left( \sum_{j=1}^{n} \frac{(1+\omega(\varpi_j))}{\sum_{j=1}^{n}(1+\omega(\varpi_j))} \left( -\log\left(1-\frac{\mathfrak{l}_j}{\partial}\right) \right)^\pi \right)^{\frac{1}{\pi}}}}, \\[20pt] \left( \left( 1 - \mathbb{e}^{-\left( \sum_{j=1}^{n} \frac{(1+\omega(\varpi_j))}{\sum_{j=1}^{n}(1+\omega(\varpi_j))} \left( -\log(1-\Xi_{\varpi_j}) \right)^\pi \right)^{\frac{1}{\pi}}} \right) + i \left( 1 - \mathbb{e}^{-\left( \sum_{j=1}^{n} \frac{(1+\omega(\varpi_j))}{\sum_{j=1}^{n}(1+\omega(\varpi_j))} \left( -\log(1-\Omega_{\varpi_j}) \right)^\pi \right)^{\frac{1}{\pi}}} \right), \\[20pt] -\left( \mathbb{e}^{-\left( \sum_{j=1}^{n} \frac{(1+\omega(\varpi_j))}{\sum_{j=1}^{n}(1+\omega(\varpi_j))} \left( -\log(|\Psi_{\varpi_j}|) \right)^\pi \right)^{\frac{1}{\pi}}} \right) + i \left( -\left( \mathbb{e}^{-\left( \sum_{j=1}^{n} \frac{(1+\omega(\varpi_j))}{\sum_{j=1}^{n}(1+\omega(\varpi_j))} \left( -\log(|\Phi_{\varpi_j}|) \right)^\pi \right)^{\frac{1}{\pi}}} \right) \right) \end{array} \right)$$

**Proof:** For the simplification of the above information, we aim to use the technique of mathematical induction. For this, if $n = 2$, thus

$$\frac{(1+\omega(\varpi_1))}{\sum_{j=1}^{2}(1+\omega(\varpi_j))} \varpi_1$$

$$= \left( \begin{array}{c} \partial^{1-\mathbb{e}^{-\left( \frac{(1+\omega(\varpi_1))}{\sum_{j=1}^{2}(1+\omega(\varpi_j))} \left( -\log\left(1-\frac{\mathfrak{l}_1}{\partial}\right) \right)^\pi \right)^{\frac{1}{\pi}}}}, \\[20pt] \left( \left( 1 - \mathbb{e}^{-\left( \frac{(1+\omega(\varpi_1))}{\sum_{j=1}^{2}(1+\omega(\varpi_j))} \left( -\log(1-\Xi_{\varpi_1}) \right)^\pi \right)^{\frac{1}{\pi}}} \right) + i \left( 1 - \mathbb{e}^{-\left( \frac{(1+\omega(\varpi_1))}{\sum_{j=1}^{2}(1+\omega(\varpi_j))} \left( -\log(1-\Omega_{\varpi_1}) \right)^\pi \right)^{\frac{1}{\pi}}} \right), \\[20pt] -\left( \mathbb{e}^{-\left( \frac{(1+\omega(\varpi_1))}{\sum_{j=1}^{2}(1+\omega(\varpi_j))} \left( -\log(|\Psi_{\varpi_1}|) \right)^\pi \right)^{\frac{1}{\pi}}} \right) + i \left( -\left( \mathbb{e}^{-\left( \frac{(1+\omega(\varpi_1))}{\sum_{j=1}^{2}(1+\omega(\varpi_j))} \left( -\log(|\Phi_{\varpi_1}|) \right)^\pi \right)^{\frac{1}{\pi}}} \right) \right) \end{array} \right)$$

$$\frac{(1+\omega(\varpi_2))}{\sum_{j=1}^{2}(1+\omega(\varpi_j))}\varpi_2^{BC} = \left( \begin{array}{c} \partial\left(1-\mathbb{C}^{-\left(\frac{(1+\omega(\varpi_2))}{\sum_{j=1}^{2}(1+\omega(\varpi_j))}\left(-\log\left(1-\frac{\mathfrak{l}_2}{\partial}\right)\right)^{\pi}\right)^{\frac{1}{\pi}}}\right), \\ \left(\left(1-\mathbb{C}^{-\left(\frac{(1+\omega(\varpi_2))}{\sum_{j=1}^{2}(1+\omega(\varpi_j))}(-\log(1-\Xi_{\varpi_2}))^{\pi}\right)^{\frac{1}{\pi}}}\right)+i\left(1-\mathbb{C}^{-\left(\frac{(1+\omega(\varpi_2))}{\sum_{j=1}^{2}(1+\omega(\varpi_j))}(-\log(1-\Omega_{\varpi_2}))^{\pi}\right)^{\frac{1}{\pi}}}\right)\right), \\ -\left(\mathbb{C}^{-\left(\frac{(1+\omega(\varpi_2))}{\sum_{j=1}^{2}(1+\omega(\varpi_j))}(-\log(|\Psi_{\varpi_2}|))^{\pi}\right)^{\frac{1}{\pi}}}\right)+i\left(-\left(\mathbb{C}^{-\left(\frac{(1+\omega(\varpi_2))}{\sum_{j=1}^{2}(1+\omega(\varpi_j))}(-\log(|\Phi_{\varpi_2}|))^{\pi}\right)^{\frac{1}{\pi}}}\right)\right) \end{array} \right)$$

Thus, by combining the above two equations, such as

$$BCFLAAPOA\left(\varpi_1^{BC},\varpi_2^{BC}\right) = \frac{(1+\omega(\varpi_1))}{\sum_{j=1}^{2}(1+\omega(\varpi_j))}\varpi_1^{BC} \oplus \frac{(1+\omega(\varpi_2))}{\sum_{j=1}^{2}(1+\omega(\varpi_j))}\varpi_2^{BC}$$

$$= \left( \begin{array}{c} \partial\left(1-\mathbb{C}^{-\left(\frac{(1+\omega(\varpi_1))}{\sum_{j=1}^{2}(1+\omega(\varpi_j))}\left(-\log\left(1-\frac{\mathfrak{l}_1}{\partial}\right)\right)^{\pi}\right)^{\frac{1}{\pi}}}\right), \\ \left(\left(1-\mathbb{C}^{-\left(\frac{(1+\omega(\varpi_1))}{\sum_{j=1}^{2}(1+\omega(\varpi_j))}(-\log(1-\Xi_{\varpi_1}))^{\pi}\right)^{\frac{1}{\pi}}}\right)+i\left(1-\mathbb{C}^{-\left(\frac{(1+\omega(\varpi_1))}{\sum_{j=1}^{2}(1+\omega(\varpi_j))}(-\log(1-\Omega_{\varpi_1}))^{\pi}\right)^{\frac{1}{\pi}}}\right)\right), \\ -\left(\mathbb{C}^{-\left(\frac{(1+\omega(\varpi_1))}{\sum_{j=1}^{2}(1+\omega(\varpi_j))}(-\log(|\Psi_{\varpi_1}|))^{\pi}\right)^{\frac{1}{\pi}}}\right)+i\left(-\left(\mathbb{C}^{-\left(\frac{(1+\omega(\varpi_1))}{\sum_{j=1}^{2}(1+\omega(\varpi_j))}(-\log(|\Phi_{\varpi_1}|))^{\pi}\right)^{\frac{1}{\pi}}}\right)\right) \end{array} \right)$$

$$\oplus \left( \begin{array}{c} \partial\left(1-\mathbb{C}^{-\left(\frac{(1+\omega(\varpi_2))}{\sum_{j=1}^{2}(1+\omega(\varpi_j))}\left(-\log\left(1-\frac{\mathfrak{l}_2}{\partial}\right)\right)^{\pi}\right)^{\frac{1}{\pi}}}\right), \\ \left(\left(1-\mathbb{C}^{-\left(\frac{(1+\omega(\varpi_2))}{\sum_{j=1}^{2}(1+\omega(\varpi_j))}(-\log(1-\Xi_{\varpi_2}))^{\pi}\right)^{\frac{1}{\pi}}}\right)+i\left(1-\mathbb{C}^{-\left(\frac{(1+\omega(\varpi_2))}{\sum_{j=1}^{2}(1+\omega(\varpi_j))}(-\log(1-\Omega_{\varpi_2}))^{\pi}\right)^{\frac{1}{\pi}}}\right)\right), \\ -\left(\mathbb{C}^{-\left(\frac{(1+\omega(\varpi_2))}{\sum_{j=1}^{2}(1+\omega(\varpi_j))}(-\log(|\Psi_{\varpi_2}|))^{\pi}\right)^{\frac{1}{\pi}}}\right)+i\left(-\left(\mathbb{C}^{-\left(\frac{(1+\omega(\varpi_2))}{\sum_{j=1}^{2}(1+\omega(\varpi_j))}(-\log(|\Phi_{\varpi_2}|))^{\pi}\right)^{\frac{1}{\pi}}}\right)\right) \end{array} \right)$$

$$
\left(
\begin{array}{c}
\mathcal{L}\left(\partial_{1-\mathbb{e}^{-\left(\sum_{j=1}^{2}\frac{(1+\omega(\varpi_j))}{\sum_{j=1}^{n}(1+\omega(\varpi_j))}\left(-\log\left(1-\frac{\mathfrak{l}_j}{\partial}\right)\right)^{\pi}\right)^{\frac{1}{\pi}}}}\right)^{,}\\[3em]
= \left(\left(1-\mathbb{e}^{-\left(\sum_{j=1}^{2}\frac{(1+\omega(\varpi_j))}{\sum_{j=1}^{n}(1+\omega(\varpi_j))}\left(-\log(1-\Xi_{\varpi_j})\right)^{\pi}\right)^{\frac{1}{\pi}}}\right)+i\left(1-\mathbb{e}^{-\left(\sum_{j=1}^{2}\frac{(1+\omega(\varpi_j))}{\sum_{j=1}^{n}(1+\omega(\varpi_j))}\left(-\log(1-\Omega_{\varpi_j})\right)^{\pi}\right)^{\frac{1}{\pi}}}\right),\\[3em]
-\left(\mathbb{e}^{-\left(\sum_{j=1}^{2}\frac{(1+\omega(\varpi_j))}{\sum_{j=1}^{n}(1+\omega(\varpi_j))}\left(-\log(|\Psi_{\varpi_j}|)\right)^{\pi}\right)^{\frac{1}{\pi}}}\right)+i\left(-\left(\mathbb{e}^{-\left(\sum_{j=1}^{2}\frac{(1+\omega(\varpi_j))}{\sum_{j=1}^{n}(1+\omega(\varpi_j))}\left(-\log(|\Phi_{\varpi_j}|)\right)^{\pi}\right)^{\frac{1}{\pi}}}\right)\right)
\end{array}
\right)
$$

Hence, the proposed theory holds for $n = 2$. Further, we assume that the proposed theory also holds for $n = n'$, such as

$BCFLAAPOA(\varpi_1, \varpi_2, \ldots, \varpi_{n'})$

$$
\left(
\begin{array}{c}
\mathcal{L}\left(\partial_{1-\mathbb{e}^{-\left(\sum_{j=1}^{n'}\frac{(1+\omega(\varpi_j))}{\sum_{j=1}^{n'}(1+\omega(\varpi_j))}\left(-\log\left(1-\frac{\mathfrak{l}_j}{\partial}\right)\right)^{\pi}\right)^{\frac{1}{\pi}}}}\right)^{,}\\[3em]
= \left(\left(1-\mathbb{e}^{-\left(\sum_{j=1}^{n'}\frac{(1+\omega(\varpi_j))}{\sum_{j=1}^{n'}(1+\omega(\varpi_j))}\left(-\log(1-\Xi_{\varpi_j})\right)^{\pi}\right)^{\frac{1}{\pi}}}\right)+i\left(1-\mathbb{e}^{-\left(\sum_{j=1}^{n'}\frac{(1+\omega(\varpi_j))}{\sum_{j=1}^{n'}(1+\omega(\varpi_j))}\left(-\log(1-\Omega_{\varpi_j})\right)^{\pi}\right)^{\frac{1}{\pi}}}\right),\\[3em]
-\left(\mathbb{e}^{-\left(\sum_{j=1}^{n'}\frac{(1+\omega(\varpi_j))}{\sum_{j=1}^{n'}(1+\omega(\varpi_j))}\left(-\log(|\Psi_{\varpi_j}|)\right)^{\pi}\right)^{\frac{1}{\pi}}}\right)+i\left(-\left(\mathbb{e}^{-\left(\sum_{j=1}^{n'}\frac{(1+\omega(\varpi_j))}{\sum_{j=1}^{n'}(1+\omega(\varpi_j))}\left(-\log(|\Phi_{\varpi_j}|)\right)^{\pi}\right)^{\frac{1}{\pi}}}\right)\right)
\end{array}
\right)
$$

Then, we proved that the proposed theory also holds for $n = n'+1$, such as

$$
BCFLAAPOA(\varpi_1^{BC}, \varpi_2^{BC}, \ldots, \varpi_{n'+1}^{BC}) = BCFLAAPOA(\varpi_1, \varpi_2, \ldots, \varpi_{n'+1})
$$

$$
= \frac{(1+\omega(\varpi_1))}{\sum_{j=1}^{n'+1}(1+\omega(\varpi_j))}\varpi_1^{BC} \oplus \frac{(1+\omega(\varpi_2))}{\sum_{j=1}^{n'+1}(1+\omega(\varpi_j))}\varpi_2^{BC} \oplus \ldots \oplus \frac{(1+\omega(\varpi_{n'+1}))}{\sum_{j=1}^{n'+1}(1+\omega(\varpi_j))}\varpi_{n'+1}^{BC}
$$

$$
= \oplus_{j=1}^{n'}\frac{(1+\omega(\varpi_j))}{\sum_{j=1}^{n'}(1+\omega(\varpi_j))}\varpi_j^{BC} \oplus \frac{(1+\omega(\varpi_{n'+1}))}{\sum_{j=1}^{n'+1}(1+\omega(\varpi_j))}\varpi_{n'+1}^{BC}
$$

$$
= \left( \begin{array}{c}
\mathcal{L}_{\partial} \left( 1 - \mathbb{C}^{-\left( \sum_{j=1}^{n'} \frac{(1+\omega(\varpi_j))}{\sum_{j=1}^{n'}(1+\omega(\varpi_j))} \left( -\log\left(1 - \frac{I_j}{\partial}\right)\right)^{\pi} \right)^{\frac{1}{\pi}}} \right), \\[2em]
\left( \left( 1 - \mathbb{C}^{-\left( \sum_{j=1}^{n'} \frac{(1+\omega(\varpi_j))}{\sum_{j=1}^{n'}(1+\omega(\varpi_j))} \left( -\log(1 - \Xi_{\varpi_j})\right)^{\pi} \right)^{\frac{1}{\pi}}} \right) + i\left( 1 - \mathbb{C}^{-\left( \sum_{j=1}^{n'} \frac{(1+\omega(\varpi_j))}{\sum_{j=1}^{n'}(1+\omega(\varpi_j))} \left( -\log(1 - \Omega_{\varpi_j})\right)^{\pi} \right)^{\frac{1}{\pi}}} \right), \\[2em]
-\left( \mathbb{C}^{-\left( \sum_{j=1}^{n'} \frac{(1+\omega(\varpi_j))}{\sum_{j=1}^{n'}(1+\omega(\varpi_j))} \left( -\log(|\Psi_{\varpi_j}|)\right)^{\pi} \right)^{\frac{1}{\pi}}} \right) + i\left( -\left( \mathbb{C}^{-\left( \sum_{j=1}^{n'} \frac{(1+\omega(\varpi_j))}{\sum_{j=1}^{n'}(1+\omega(\varpi_j))} \left( -\log(|\Phi_{\varpi_j}|)\right)^{\pi} \right)^{\frac{1}{\pi}}} \right) \right) \right)
\end{array} \right)
$$

$$
\oplus \frac{(1+\omega(\varpi_{n'+1}))}{\sum_{j=1}^{n'+1}(1+\omega(\varpi_j))} \varpi_{n'+1}^{BC}
$$

$$
= \left( \begin{array}{c}
\mathcal{L}_{\partial} \left( 1 - \mathbb{C}^{-\left( \sum_{j=1}^{n'} \frac{(1+\omega(\varpi_j))}{\sum_{j=1}^{n'}(1+\omega(\varpi_j))} \left( -\log\left(1 - \frac{I_j}{\partial}\right)\right)^{\pi} \right)^{\frac{1}{\pi}}} \right), \\[2em]
\left( \left( 1 - \mathbb{C}^{-\left( \sum_{j=1}^{n'} \frac{(1+\omega(\varpi_j))}{\sum_{j=1}^{n'}(1+\omega(\varpi_j))} \left( -\log(1 - \Xi_{\varpi_j})\right)^{\pi} \right)^{\frac{1}{\pi}}} \right) + i\left( 1 - \mathbb{C}^{-\left( \sum_{j=1}^{n'} \frac{(1+\omega(\varpi_j))}{\sum_{j=1}^{n'}(1+\omega(\varpi_j))} \left( -\log(1 - \Omega_{\varpi_j})\right)^{\pi} \right)^{\frac{1}{\pi}}} \right), \\[2em]
-\left( \mathbb{C}^{-\left( \sum_{j=1}^{n'} \frac{(1+\omega(\varpi_j))}{\sum_{j=1}^{n'}(1+\omega(\varpi_j))} \left( -\log(|\Psi_{\varpi_j}|)\right)^{\pi} \right)^{\frac{1}{\pi}}} \right) + i\left( -\left( \mathbb{C}^{-\left( \sum_{j=1}^{n'} \frac{(1+\omega(\varpi_j))}{\sum_{j=1}^{n'}(1+\omega(\varpi_j))} \left( -\log(|\Phi_{\varpi_j}|)\right)^{\pi} \right)^{\frac{1}{\pi}}} \right) \right) \right)
\end{array} \right)
$$

$$
\oplus \left( \begin{array}{c}
\mathcal{L}_{\partial} \left( 1 - \mathbb{C}^{-\left( \frac{(1+\omega(\varpi_{n'+1}))}{\sum_{j=1}^{n'+1}(1+\omega(\varpi_j))} \left( -\log\left(1 - \frac{I_{n'+1}}{\partial}\right)\right)^{\pi} \right)^{\frac{1}{\pi}}} \right), \\[2em]
\left( \left( 1 - \mathbb{C}^{-\left( \frac{(1+\omega(\varpi_{n'+1}))}{\sum_{j=1}^{n'+1}(1+\omega(\varpi_j))} \left( -\log(1 - \Xi_{\varpi_{n'+1}})\right)^{\pi} \right)^{\frac{1}{\pi}}} \right) + i\left( 1 - \mathbb{C}^{-\left( \frac{(1+\omega(\varpi_{n'+1}))}{\sum_{j=1}^{n'+1}(1+\omega(\varpi_j))} \left( -\log(1 - \Omega_{\varpi_{n'+1}})\right)^{\pi} \right)^{\frac{1}{\pi}}} \right), \\[2em]
-\left( \mathbb{C}^{-\left( \frac{(1+\omega(\varpi_{n'+1}))}{\sum_{j=1}^{n'+1}(1+\omega(\varpi_j))} \left( -\log(|\Psi_{\varpi_{n'+1}}|)\right)^{\pi} \right)^{\frac{1}{\pi}}} \right) + i\left( -\left( \mathbb{C}^{-\left( \frac{(1+\omega(\varpi_{n'+1}))}{\sum_{j=1}^{n'+1}(1+\omega(\varpi_j))} \left( -\log(|\Phi_{\varpi_{n'+1}}|)\right)^{\pi} \right)^{\frac{1}{\pi}}} \right) \right) \right)
\end{array} \right)
$$

$$
\begin{pmatrix}
\mathcal{L} \left( {}_{\partial} 1 - \mathbb{e}^{-\left( \sum_{j=1}^{n'+1} \frac{(1+\omega(\varpi_j))}{\sum_{j=1}^{n'+1}(1+\omega(\varpi_j))} \left( -\log\left( 1 - \frac{\mathfrak{l}_j}{\partial} \right) \right)^{\pi} \right)^{\frac{1}{\pi}}} \right)
\end{pmatrix}
,
$$

$$
=
\begin{pmatrix}
\left( 1 - \mathbb{e}^{-\left( \sum_{j=1}^{n'+1} \frac{(1+\omega(\varpi_j))}{\sum_{j=1}^{n'+1}(1+\omega(\varpi_j))} \left( -\log(1 - \Xi_{\varpi_j}) \right)^{\pi} \right)^{\frac{1}{\pi}}} \right) + i \left( 1 - \mathbb{e}^{-\left( \sum_{j=1}^{n'+1} \frac{(1+\omega(\varpi_j))}{\sum_{j=1}^{n'+1}(1+\omega(\varpi_j))} \left( -\log(1 - \Omega_{\varpi_j}) \right)^{\pi} \right)^{\frac{1}{\pi}}} \right),
\\[2em]
- \left( \mathbb{e}^{-\left( \sum_{j=1}^{n'+1} \frac{(1+\omega(\varpi_j))}{\sum_{j=1}^{n'+1}(1+\omega(\varpi_j))} \left( -\log(|\Psi_{\varpi_j}|) \right)^{\pi} \right)^{\frac{1}{\pi}}} \right) + i \left( - \left( \mathbb{e}^{-\left( \sum_{j=1}^{n'+1} \frac{(1+\omega(\varpi_j))}{\sum_{j=1}^{n'+1}(1+\omega(\varpi_j))} \left( -\log(|\Phi_{\varpi_j}|) \right)^{\pi} \right)^{\frac{1}{\pi}}} \right) \right)
\end{pmatrix}
$$

Hence, the proposed theory holds for $n = n'+1$, it means that the proposed theory holds for non-negative integers. Moreover, we simplify some basic or major properties for the above operators, called idempotency, monotonicity, and boundedness.

**Property 1:** Let $\varpi_j^{BC} = (\mathcal{L}_{\mathfrak{l}_j}, \Xi_{\varpi_j} + i\Omega_{\varpi_j}, \Psi_{\varpi_j} + i\Phi_{\varpi_j}), j = 1, 2, \ldots, n$ be the collection of BCFLNs. Then,

1. Idempotency: When $\varpi_j = \varpi, j = 1, 2, \ldots, n$, thus

$$
BCFLAAPOA(\varpi_1, \varpi_2, \ldots, \varpi_n) = \varpi
$$

2. Monotonicity: When $\varpi_j \leq \varpi_j^{@}, j = 1, 2, \ldots, n$, thus

$$
BCFLAAPOA(\varpi_1, \varpi_2, \ldots, \varpi_n) \leq BCFLAAPOA(\varpi_1^{@}, \varpi_2^{@}, \ldots, \varpi_n^{@})
$$

3. Boundedness: When $\varpi_j^- = \min_j(\varpi_j)$ and $\varpi_j^+ = \max_j(\varpi_j)$, thus

$$
\varpi_j^- \leq BCFLAAPOA(\varpi_1, \varpi_2, \ldots, \varpi_n) \leq \varpi_j^+
$$

**Proof:** By using the information in Def. (4) and Def. (5), we prove the required results.

1. When $\varpi_j = \varpi, j = 1, 2, \ldots, n$, thus

$$BCFLAAPOA(\varpi_1, \varpi_2, \ldots, \varpi_n)$$

$$= \left( \begin{array}{c} \mathcal{L}\left( \partial\left( 1 - e^{-\left( \sum_{j=1}^n \frac{(1+\omega(\varpi_j))}{\sum_{j=1}^n (1+\omega(\varpi_j))} \left(-\log\left(1-\frac{l_j}{\partial}\right)\right)^\pi \right)^{\frac{1}{\pi}}} \right)' \right) \\ \left( \left( 1 - e^{-\left( \sum_{j=1}^n \frac{(1+\omega(\varpi_j))}{\sum_{j=1}^n (1+\omega(\varpi_j))} (-\log(1-\Xi_{\varpi_j}))^\pi \right)^{\frac{1}{\pi}}} \right) + i\left( 1 - e^{-\left( \sum_{j=1}^n \frac{(1+\omega(\varpi_j))}{\sum_{j=1}^n (1+\omega(\varpi_j))} (-\log(1-\Omega_{\varpi_j}))^\pi \right)^{\frac{1}{\pi}}} \right), \\ -\left( e^{-\left( \sum_{j=1}^n \frac{(1+\omega(\varpi_j))}{\sum_{j=1}^n (1+\omega(\varpi_j))} (-\log(|\Psi_{\varpi_j}|))^\pi \right)^{\frac{1}{\pi}}} \right) + i\left( -\left( e^{-\left( \sum_{j=1}^n \frac{(1+\omega(\varpi_j))}{\sum_{j=1}^n (1+\omega(\varpi_j))} (-\log(|\Phi_{\varpi_j}|))^\pi \right)^{\frac{1}{\pi}}} \right) \right) \end{array} \right)$$

$$= \left( \begin{array}{c} \mathcal{L}\left( \partial\left( 1 - e^{-\left( \frac{\sum_{j=1}^n (1+\omega(\varpi_j))}{\sum_{j=1}^n (1+\omega(\varpi_j))} \left(-\log\left(1-\frac{l}{\partial}\right)\right)^\pi \right)^{\frac{1}{\pi}}} \right)' \right) \\ \left( \left( 1 - e^{-\left( \frac{\sum_{j=1}^n (1+\omega(\varpi_j))}{\sum_{j=1}^n (1+\omega(\varpi_j))} (-\log(1-\Xi_{\varpi}))^\pi \right)^{\frac{1}{\pi}}} \right) + i\left( 1 - e^{-\left( \frac{\sum_{j=1}^n (1+\omega(\varpi_j))}{\sum_{j=1}^n (1+\omega(\varpi_j))} (-\log(1-\Omega_{\varpi}))^\pi \right)^{\frac{1}{\pi}}} \right), \\ -\left( e^{-\left( \frac{\sum_{j=1}^n (1+\omega(\varpi_j))}{\sum_{j=1}^n (1+\omega(\varpi_j))} (-\log(|\Psi_{\varpi}|))^\pi \right)^{\frac{1}{\pi}}} \right) + i\left( -\left( e^{-\left( \frac{\sum_{j=1}^n (1+\omega(\varpi_j))}{\sum_{j=1}^n (1+\omega(\varpi_j))} (-\log(|\Phi_{\varpi}|))^\pi \right)^{\frac{1}{\pi}}} \right) \right) \end{array} \right)$$

$$= \begin{pmatrix} \mathcal{L}_{\partial\left(1-\mathbb{e}^{-\left(\left(-\log\left(1-\frac{\mathfrak{l}}{\partial}\right)\right)^{\pi}\right)^{\frac{1}{\pi}}}\right)}, \\ \left(1-\mathbb{e}^{-\left((-\log(1-\Xi_{\varpi}))^{\pi}\right)^{\frac{1}{\pi}}}\right) + i\left(1-\mathbb{e}^{-\left((-\log(1-\Omega_{\varpi}))^{\pi}\right)^{\frac{1}{\pi}}}\right), \\ -\left(\mathbb{e}^{-\left((-\log(|\Psi_{\varpi}|))^{\pi}\right)^{\frac{1}{\pi}}}\right) + i\left(-\left(\mathbb{e}^{-\left((-\log(|\Phi_{\varpi}|))^{\pi}\right)^{\frac{1}{\pi}}}\right)\right) \end{pmatrix}$$

$$= \begin{pmatrix} \mathcal{L}_{\partial\left(1-\mathbb{e}^{\log\left(1-\frac{\mathfrak{l}}{\partial}\right)}\right)}, \\ \left(1-\mathbb{e}^{\log(1-\Xi_{\varpi})}\right) + i\left(1-\mathbb{e}^{\log(1-\Omega_{\varpi})}\right), \\ -\left(\mathbb{e}^{\log(|\Psi_{\varpi}|)}\right) + i\left(-\left(\mathbb{e}^{\log(|\Phi_{\varpi}|)}\right)\right) \end{pmatrix} = \begin{pmatrix} \mathcal{L}_{\mathfrak{l}}, \\ \Xi_{\varpi} + i\Omega_{\varpi}, \\ \Psi_{\varpi} + i\Phi_{\varpi} \end{pmatrix} = \varpi.$$

2. When $\varpi_j \leq \varpi_j^{@}, j = 1, 2, \ldots, n$, thus $\mathfrak{l}_j \leq \mathfrak{l}_j^{@}, \Xi_{\varpi_j} \leq \Xi_{\varpi_j}^{@}, \Omega_{\varpi_j} \leq \Omega_{\varpi_j}^{@}$ and $\Psi_{\varpi_j} \leq \Psi_{\varpi_j}^{@}, \Phi_{\varpi_j} \leq \Phi_{\varpi_j}^{@}$, then

$$\mathfrak{l}_j \leq \mathfrak{l}_j^{@} \Rightarrow -\frac{\mathfrak{l}_j}{\partial} \geq -\frac{\mathfrak{l}_j^{@}}{\partial} \Rightarrow 1-\frac{\mathfrak{l}_j}{\partial} \geq 1-\frac{\mathfrak{l}_j^{@}}{\partial}$$

$$\Rightarrow -\log\left(1-\frac{\mathfrak{l}_j}{\partial}\right) \leq -\log\left(1-\frac{\mathfrak{l}_j^{@}}{\partial}\right)$$

$$\Rightarrow \frac{(1+\omega(\varpi_j))}{\sum_{j=1}^{n}(1+\omega(\varpi_j))}\left(-\log\left(1-\frac{\mathfrak{l}_j}{\partial}\right)\right)^{\pi} \leq \frac{(1+\omega(\varpi_j))}{\sum_{j=1}^{n}(1+\omega(\varpi_j))}\left(-\log\left(1-\frac{\mathfrak{l}_j^{@}}{\partial}\right)\right)^{\pi}$$

$$\Rightarrow -\left(\sum_{j=1}^{n}\frac{(1+\omega(\varpi_j))}{\sum_{j=1}^{n}(1+\omega(\varpi_j))}\left(-\log\left(1-\frac{\mathfrak{l}_j}{\partial}\right)\right)^{\pi}\right)^{\frac{1}{\pi}} \geq -\left(\sum_{j=1}^{n}\frac{(1+\omega(\varpi_j))}{\sum_{j=1}^{n}(1+\omega(\varpi_j))}\left(-\log\left(1-\frac{\mathfrak{l}_j^{@}}{\partial}\right)\right)^{\pi}\right)^{\frac{1}{\pi}}$$

$$\Rightarrow -\mathbb{e}^{-\left(\sum_{j=1}^{n}\frac{(1+\omega(\varpi_j))}{\sum_{j=1}^{n}(1+\omega(\varpi_j))}\left(-\log\left(1-\frac{\mathfrak{l}_j}{\partial}\right)\right)^{\pi}\right)^{\frac{1}{\pi}}} \leq -\mathbb{e}^{-\left(\sum_{j=1}^{n}\frac{(1+\omega(\varpi_j))}{\sum_{j=1}^{n}(1+\omega(\varpi_j))}\left(-\log\left(1-\frac{\mathfrak{l}_j^{@}}{\partial}\right)\right)^{\pi}\right)^{\frac{1}{\pi}}}$$

$$\Rightarrow \partial \left( 1 - \mathbb{e}^{-\left( \sum_{j=1}^{n} \frac{(1+\omega(\varpi_j))}{\sum_{j=1}^{n}(1+\omega(\varpi_j))} \left( -\log\left(1-\frac{l_j}{\partial}\right)\right)^{\pi}\right)^{\frac{1}{\pi}}} \right) \leq \partial \left( 1 - \mathbb{e}^{-\left( \sum_{j=1}^{n} \frac{(1+\omega(\varpi_j))}{\sum_{j=1}^{n}(1+\omega(\varpi_j))} \left( -\log\left(1-\frac{l_j^{@}}{\partial}\right)\right)^{\pi}\right)^{\frac{1}{\pi}}} \right)$$

Hence, we concluded that.

$$\Rightarrow \mathcal{L}_{\partial} \left( 1 - \mathbb{e}^{-\left( \sum_{j=1}^{n} \frac{(1+\omega(\varpi_j))}{\sum_{j=1}^{n}(1+\omega(\varpi_j))} \left( -\log\left(1-\frac{l_j}{\partial}\right)\right)^{\pi}\right)^{\frac{1}{\pi}}} \right) \leq \mathcal{L}_{\partial} \left( 1 - \mathbb{e}^{-\left( \sum_{j=1}^{n} \frac{(1+\omega(\varpi_j))}{\sum_{j=1}^{n}(1+\omega(\varpi_j))} \left( -\log\left(1-\frac{l_j^{@}}{\partial}\right)\right)^{\pi}\right)^{\frac{1}{\pi}}} \right)$$

Further, we simplify the positive membership function, such as:

$$\Xi_{\varpi_j} \leq \Xi_{\varpi_j}^{@} \Rightarrow 1 - \Xi_{\varpi_j} \geq 1 - \Xi_{\varpi_j}^{@}$$

$$\Rightarrow (-\log(1 - \Xi_{\varpi_j}))^{\pi} \leq (-\log(1 - \Xi_{\varpi_j}^{@}))^{\pi}$$

$$\Rightarrow \sum_{j=1}^{n} \frac{(1+\omega(\varpi_j))}{\sum_{j=1}^{n}(1+\omega(\varpi_j))} (-\log(1 - \Xi_{\varpi_j}))^{\pi} \leq \sum_{j=1}^{n} \frac{(1+\omega(\varpi_j))}{\sum_{j=1}^{n}(1+\omega(\varpi_j))} (-\log(1 - \Xi_{\varpi_j}^{@}))^{\pi}$$

$$\Rightarrow -\left( \sum_{j=1}^{n} \frac{(1+\omega(\varpi_j))}{\sum_{j=1}^{n}(1+\omega(\varpi_j))} (-\log(1 - \Xi_{\varpi_j}))^{\pi} \right)^{\frac{1}{\pi}} \geq -\left( \sum_{j=1}^{n} \frac{(1+\omega(\varpi_j))}{\sum_{j=1}^{n}(1+\omega(\varpi_j))} (-\log(1 - \Xi_{\varpi_j}^{@}))^{\pi} \right)^{\frac{1}{\pi}}$$

$$\Rightarrow 1 - \mathbb{e}^{-\left( \sum_{j=1}^{n} \frac{(1+\omega(\varpi_j))}{\sum_{j=1}^{n}(1+\omega(\varpi_j))} (-\log(1-\Xi_{\varpi_j}))^{\pi} \right)^{\frac{1}{\pi}}} \leq 1 - \mathbb{e}^{-\left( \sum_{j=1}^{n} \frac{(1+\omega(\varpi_j))}{\sum_{j=1}^{n}(1+\omega(\varpi_j))} (-\log(1-\Xi_{\varpi_j}^{@}))^{\pi} \right)^{\frac{1}{\pi}}}$$

Similarly, we evaluate the imaginary part for positive membership function, such as

$$\Omega_{\varpi_j} \leq \Omega_{\varpi_j}^{@} \Rightarrow 1 - \mathbb{e}^{-\left( \sum_{j=1}^{n} \frac{(1+\omega(\varpi_j))}{\sum_{j=1}^{n}(1+\omega(\varpi_j))} (-\log(1-\Omega_{\varpi_j}))^{\pi} \right)^{\frac{1}{\pi}}} \leq 1 - \mathbb{e}^{-\left( \sum_{j=1}^{n} \frac{(1+\omega(\varpi_j))}{\sum_{j=1}^{n}(1+\omega(\varpi_j))} (-\log(1-\Omega_{\varpi_j}^{@}))^{\pi} \right)^{\frac{1}{\pi}}}$$

Further, we determine the negative membership function, such as

$$\Psi_{\varpi_j} \leq \Psi_{\varpi_j}^{@} \Rightarrow |\Psi_{\varpi_j}| \geq |\Psi_{\varpi_j}^{@}| \Rightarrow (-\log(|\Psi_{\varpi_j}|))^{\pi} \geq (-\log(|\Psi_{\varpi_j}^{@}|))^{\pi}$$

$$\Rightarrow \sum_{j=1}^{n} \frac{(1+\omega(\varpi_j))}{\sum_{j=1}^{n}(1+\omega(\varpi_j))} (-\log(|\Psi_{\varpi_j}|))^{\pi} \geq \sum_{j=1}^{n} \frac{(1+\omega(\varpi_j))}{\sum_{j=1}^{n}(1+\omega(\varpi_j))} (-\log(|\Psi_{\varpi_j}^{@}|))^{\pi}$$

$$\Rightarrow -\left(\mathbb{e}^{-\left(\sum_{j=1}^{n}\frac{(1+\omega(\varpi_j))}{\sum_{j=1}^{n}(1+\omega(\varpi_j))}(-\log(|\Psi_{\varpi_j}|))^{\pi}\right)^{\frac{1}{\pi}}}\right) \leq -\left(\mathbb{e}^{-\left(\sum_{j=1}^{n}\frac{(1+\omega(\varpi_j))}{\sum_{j=1}^{n}(1+\omega(\varpi_j))}(-\log(|\Psi_{\varpi_j}^{@}|))^{\pi}\right)^{\frac{1}{\pi}}}\right)$$

Similarly, we evaluate the imaginary part for negative membership function, such as

$$\Phi_{\varpi_j} \leq \Phi_{\varpi_j}^{@} \Rightarrow -\left(\mathbb{e}^{-\left(\sum_{j=1}^{n}\frac{(1+\omega(\varpi_j))}{\sum_{j=1}^{n}(1+\omega(\varpi_j))}(-\log(|\Phi_{\varpi_j}|))^{\pi}\right)^{\frac{1}{\pi}}}\right) \leq -\left(\mathbb{e}^{-\left(\sum_{j=1}^{n}\frac{(1+\omega(\varpi_j))}{\sum_{j=1}^{n}(1+\omega(\varpi_j))}(-\log(|\Phi_{\varpi_j}^{@}|))^{\pi}\right)^{\frac{1}{\pi}}}\right)$$

Finally, by using the technique of score values, we can easily derive the required result, such as

$$BCFLAAPOA(\varpi_1, \varpi_2, \ldots, \varpi_n) \leq BCFLAAPOA(\varpi_1^{@}, \varpi_2^{@}, \ldots, \varpi_n^{@}).$$

3. When $\varpi_j^- = \min_j(\varpi_j)$ and $\varpi_j^+ = \max_j(\varpi_j)$, thus

$$\varpi_j^- = \min_j(\varpi_j) \leq (\varpi_1, \varpi_2, \ldots, \varpi_n)$$

$$\varpi_j^+ = \max_j(\varpi_j) \geq (\varpi_1, \varpi_2, \ldots, \varpi_n)$$

thus

$$\varpi_j^- \leq BCFLAAPOA(\varpi_1, \varpi_2, \ldots, \varpi_n) \leq \varpi_j^+.$$

**Definition 6:** For any finite family of BCFLNs $\varpi_j^{BC} = (\mathcal{L}_{l_j}, \Xi_{\varpi_j} + i\Omega_{\varpi_j}, \Psi_{\varpi_j} + i\Phi_{\varpi_j}), j = 1, 2, \ldots, n$, the model of the BCFLAAPOWA operator is described and illustrated below:

$$BCFLAAPOWA : Z^n \rightarrow Z$$

by

$$BCFLAAPOWA(\varpi_1, \varpi_2, \ldots, \varpi_n)$$
$$= \frac{\Psi_1(1 + \omega(\varpi_1))}{\sum_{j=1}^{n} \Psi_j(1 + \omega(\varpi_j))}\varpi_1 \oplus \frac{\Psi_2(1 + \omega(\varpi_2))}{\sum_{j=1}^{n} \Psi_j(1 + \omega(\varpi_j))}\varpi_2 \oplus \ldots \oplus \frac{\Psi_n(1 + \omega(\varpi_n))}{\sum_{j=1}^{n} \Psi_j(1 + \omega(\varpi_j))}\varpi_n$$
$$= \oplus_{j=1}^{n} \frac{\Psi_j(1 + \omega(\varpi_j))}{\sum_{j=1}^{n} \Psi_j(1 + \omega(\varpi_j))}\varpi_j$$

Where $\omega(\varpi_j) = \sum_{j \neq k=1}^{n} SP(\varpi_j, \varpi_k)$ and $SP(\varpi_j, \varpi_k) = 1 - DS(\varpi_j, \varpi_k)$, with some conditions:

1. $SP(\varpi_j, \varpi_k) \in [0, 1]$.

2. $SP(\varpi_j, \varpi_k) = SP(\varpi_k, \varpi_j)$.

3. If $SP(\varpi_j, \varpi_k) < SP(\varpi_l, \varpi_m)$ then $DS(\varpi_j, \varpi_k) \geq DS(\varpi_l, \varpi_m)$.

Further,

$$DS\left(\varpi_j, \varpi_k\right) = \frac{1}{2}\left(\frac{|l_j - l_k|}{\partial} + \frac{1}{4}\left(|\Xi_{\varpi_j} - \Xi_{\varpi_k}| + |\Omega_{\varpi_j} - \Omega_{\varpi_k}| + |\Psi_{\varpi_j} - \Psi_{\varpi_k}| + |\Phi_{\varpi_j} - \Phi_{\varpi_k}|\right)\right)$$

Where the weight vector is represented by $\Psi_j \in [0,1]$ with a condition that is $\sum_{j=1}^{n} \Psi_j = 1$. The term $Z^n$ contained the collection of BCFLNs. Further, by using the information in Def. (4), we aim to calculate the aggregated values of the information in Def. (6).

**Theorem 2:** Let $\varpi_j^{BC} = (\mathcal{L}_{l_j}, \Xi_{\varpi_j} + i\Omega_{\varpi_j}, \Psi_{\varpi_j} + i\Phi_{\varpi_j}), j = 1, 2, \ldots, n$ be the collection of BCFLNs. Then, by using the information in Def. (4) and Def. (6), we proved that their aggregated value is also a BCFLN, such as

$BCFLAAPOWA(\varpi_1, \varpi_2, \ldots, \varpi_n)$

$$= \left( \mathcal{L}_{\partial\left(1 - \mathbb{e}^{-\left(\sum_{j=1}^{n} \frac{\Psi_j(1+\omega(\varpi_j))}{\sum_{j=1}^{n}\Psi_j(1+\omega(\varpi_j))}\left(-\log\left(1-\frac{l_j}{\partial}\right)\right)^\pi\right)^{\frac{1}{\pi}}}\right)}, \right.$$

$$\left(1 - \mathbb{e}^{-\left(\sum_{j=1}^{n} \frac{\Psi_j(1+\omega(\varpi_j))}{\sum_{j=1}^{n}\Psi_j(1+\omega(\varpi_j))}(-\log(1-\Xi_{\varpi_j}))^\pi\right)^{\frac{1}{\pi}}}\right) + i\left(1 - \mathbb{e}^{-\left(\sum_{j=1}^{n} \frac{\Psi_j(1+\omega(\varpi_j))}{\sum_{j=1}^{n}\Psi_j(1+\omega(\varpi_j))}(-\log(1-\Omega_{\varpi_j}))^\pi\right)^{\frac{1}{\pi}}}\right),$$

$$\left. -\left(\mathbb{e}^{-\left(\sum_{j=1}^{n} \frac{\Psi_j(1+\omega(\varpi_j))}{\sum_{j=1}^{n}\Psi_j(1+\omega(\varpi_j))}(-\log(|\Psi_{\varpi_j}|))^\pi\right)^{\frac{1}{\pi}}}\right) + i\left(-\mathbb{e}^{-\left(\sum_{j=1}^{n} \frac{\Psi_j(1+\omega(\varpi_j))}{\sum_{j=1}^{n}\Psi_j(1+\omega(\varpi_j))}(-\log(|\Phi_{\varpi_j}|))^\pi\right)^{\frac{1}{\pi}}}\right)\right)$$

**Proof:** The proof of this Theorem is similar to the proof of Theorem 1. Moreover, we simplify some basic or major properties for the above operators, called idempotency, monotonicity, and boundedness.

**Property 2:** Let $\varpi_j^{BC} = (\mathcal{L}_{l_j}, \Xi_{\varpi_j} + i\Omega_{\varpi_j}, \Psi_{\varpi_j} + i\Phi_{\varpi_j}), j = 1, 2, \ldots, n$ be the collection of BCFLNs. Then,

1. Idempotency: When $\varpi_j = \varpi, j = 1, 2, \ldots, n$, thus

$$BCFLAAPOWA(\varpi_1, \varpi_2, \ldots, \varpi_n) = \varpi$$

2. Monotonicity: When $\varpi_j \leq \varpi_j^{@}, j = 1, 2, \ldots, n$, thus

$$BCFLAAPOWA(\varpi_1, \varpi_2, \ldots, \varpi_n) \leq BCFLAAPOWA(\varpi_1^{@}, \varpi_2^{@}, \ldots, \varpi_n^{@})$$

3. Boundedness: When $\varpi_j^- = \min_j(\varpi_j)$ and $\varpi_j^+ = \max_j(\varpi_j)$, thus

$$\varpi_j^- \leq BCFLAAPOWA(\varpi_1, \varpi_2, \ldots, \varpi_n) \leq \varpi_j^+$$

**Proof:** The proof of this Property is similar to the proof of Property 1.

**Definition 7:** For any finite family of BCFLNs
$\varpi_j^{BC} = (\mathcal{L}_{\mathfrak{l}_j}, \Xi_{\varpi_j} + i\Omega_{\varpi_j}, \Psi_{\varpi_j} + i\Phi_{\varpi_j}), j = 1, 2, \ldots, n$, the model of the BCFLAAPOG operator
is described and illustrated below:

$$BCFLAAPOG : Z^n \rightarrow Z$$

by

$$BCFLAAPOG(\varpi_1, \varpi_2, \ldots, \varpi_n)$$
$$= \varpi_1^{\frac{(1+\omega(\varpi_1))}{\sum_{j=1}^n (1+\omega(\varpi_j))}} \otimes \varpi_2^{\frac{(1+\omega(\varpi_2))}{\sum_{j=1}^n (1+\omega(\varpi_j))}} \otimes \varpi_3^{\frac{(1+\omega(\varpi_3))}{\sum_{j=1}^n (1+\omega(\varpi_j))}} \otimes \ldots \otimes \varpi_n^{\frac{(1+\omega(\varpi_n))}{\sum_{j=1}^n (1+\omega(\varpi_j))}}$$
$$= \otimes_{j=1}^n (\varpi_j)^{\frac{(1+\omega(\varpi_j))}{\sum_{j=1}^n (1+\omega(\varpi_j))}}$$

Where $\omega(\varpi_j) = \sum_{j \neq k=1}^n SP(\varpi_j, \varpi_k)$ and $SP(\varpi_j, \varpi_k) = 1 - DS(\varpi_j, \varpi_k)$, with some
conditions:

1. $SP(\varpi_j, \varpi_k) \in [0, 1]$.

2. $SP(\varpi_j, \varpi_k) = SP(\varpi_k, \varpi_j)$.

3. If $SP(\varpi_j, \varpi_k) < SP(\varpi_l, \varpi_m)$ then $DS(\varpi_j, \varpi_k) \geq DS(\varpi_l, \varpi_m)$.

Further,

$$DS(\varpi_j, \varpi_k) = \frac{1}{2} \left( \frac{|\mathfrak{l}_j - \mathfrak{l}_k|}{\partial} + \frac{1}{4} \left( |\Xi_{\varpi_j} - \Xi_{\varpi_k}| + |\Omega_{\varpi_j} - \Omega_{\varpi_k}| + |\Psi_{\varpi_j} - \Psi_{\varpi_k}| + |\Phi_{\varpi_j} - \Phi_{\varpi_k}| \right) \right)$$

Where the weight vector is represented by $\Psi_j \in [0,1]$ with a condition that is $\sum_{j=1}^n \Psi_j = 1$.
The term $Z^n$ contained the collection of BCFLNs. Further, by using the information in Def.
(4), we aim to calculate the aggregated values of the information in Def. (7).

**Theorem 3:** Let $\varpi_j^{BC} = (\mathcal{L}_{\mathfrak{l}_j}, \Xi_{\varpi_j} + i\Omega_{\varpi_j}, \Psi_{\varpi_j} + i\Phi_{\varpi_j}), j = 1, 2, \ldots, n$ be the collection of

BCFLNs. Then, by using the information in Def. (4) and Def. (7), we proved that their

aggregated value is also a BCFLN, such as

$BCFLAAPOG(\varpi_1, \varpi_2, \ldots, \varpi_n)$

$$
= \left( \begin{array}{c} \mathcal{L}_{\partial} \left( \mathbb{e}^{-\left( \sum_{j=1}^{n} \frac{(1+\omega(\varpi_j))}{\sum_{j=1}^{n}(1+\omega(\varpi_j))} \left(-\log\left(\frac{l_j}{\partial}\right)\right)^{\pi} \right)^{\frac{1}{\pi}}} \right), \\[4em] \left( \left( \mathbb{e}^{-\left( \sum_{j=1}^{n} \frac{(1+\omega(\varpi_j))}{\sum_{j=1}^{n}(1+\omega(\varpi_j))} \left(-\log(\Xi_{\varpi_j})\right)^{\pi} \right)^{\frac{1}{\pi}}} \right) + i \left( \mathbb{e}^{-\left( \sum_{j=1}^{n} \frac{(1+\omega(\varpi_j))}{\sum_{j=1}^{n}(1+\omega(\varpi_j))} \left(-\log(\Omega_{\varpi_j})\right)^{\pi} \right)^{\frac{1}{\pi}}} \right), \\[4em] -1 + \left( \mathbb{e}^{-\left( \sum_{j=1}^{n} \frac{(1+\omega(\varpi_j))}{\sum_{j=1}^{n}(1+\omega(\varpi_j))} \left(-\log(1+\Psi_{\varpi_j})\right)^{\pi} \right)^{\frac{1}{\pi}}} \right) + i \left( -1 + \left( \mathbb{e}^{-\left( \sum_{j=1}^{n} \frac{(1+\omega(\varpi_j))}{\sum_{j=1}^{n}(1+\omega(\varpi_j))} \left(-\log(1+\Phi_{\varpi_j})\right)^{\pi} \right)^{\frac{1}{\pi}}} \right) \right) \right) \end{array} \right)
$$

**Proof:** For the simplification of the above information, we aim to use the technique of mathematical induction. For this, if $n = 2$, thus

$$
(\varpi_1)^{\frac{(1+\omega(\varpi_1))}{\sum_{j=1}^{2}(1+\omega(\varpi_j))}} = \left( \begin{array}{c} \mathcal{L}_{\partial} \left( \mathbb{e}^{-\left( \frac{(1+\omega(\varpi_1))}{\sum_{j=1}^{2}(1+\omega(\varpi_j))} \left(-\log\left(\frac{l_1}{\partial}\right)\right)^{\pi} \right)^{\frac{1}{\pi}}} \right), \\[4em] \left( \left( \mathbb{e}^{-\left( \frac{(1+\omega(\varpi_1))}{\sum_{j=1}^{2}(1+\omega(\varpi_j))} \left(-\log(\Xi_{\varpi_1})\right)^{\pi} \right)^{\frac{1}{\pi}}} \right) + i \left( \mathbb{e}^{-\left( \frac{(1+\omega(\varpi_1))}{\sum_{j=1}^{2}(1+\omega(\varpi_j))} \left(-\log(\Omega_{\varpi_1})\right)^{\pi} \right)^{\frac{1}{\pi}}} \right), \\[4em] -1 + \left( \mathbb{e}^{-\left( \frac{(1+\omega(\varpi_1))}{\sum_{j=1}^{2}(1+\omega(\varpi_j))} \left(-\log(1+\Psi_{\varpi_1})\right)^{\pi} \right)^{\frac{1}{\pi}}} \right) + i \left( -1 + \left( \mathbb{e}^{-\left( \frac{(1+\omega(\varpi_1))}{\sum_{j=1}^{2}(1+\omega(\varpi_j))} \left(-\log(1+\Phi_{\varpi_1})\right)^{\pi} \right)^{\frac{1}{\pi}}} \right) \right) \right) \end{array} \right)
$$

$$
\left(\varpi_2^{BC}\right)^{\overline{\frac{(1+\omega(\varpi_2))}{\sum_{j=1}^2 (1+\omega(\varpi_j))}}} = \left(
\begin{array}{c}
\mathcal{L}_\partial\left(\mathbb{e}^{-\left(\frac{(1+\omega(\varpi_2))}{\sum_{j=1}^2 (1+\omega(\varpi_j))}\left(-\log\left(\frac{\mathfrak{l}_2}{\partial}\right)\right)^\pi\right)^{\frac{1}{\pi}}}\right), \\[2em]
\left(\left(\mathbb{e}^{-\left(\frac{(1+\omega(\varpi_2))}{\sum_{j=1}^2 (1+\omega(\varpi_j))}(-\log(\Xi_{\varpi_2}))^\pi\right)^{\frac{1}{\pi}}}\right) + i\left(\mathbb{e}^{-\left(\frac{(1+\omega(\varpi_2))}{\sum_{j=1}^2 (1+\omega(\varpi_j))}(-\log(\Omega_{\varpi_2}))^\pi\right)^{\frac{1}{\pi}}}\right)\right), \\[2em]
-1 + \left(\mathbb{e}^{-\left(\frac{(1+\omega(\varpi_2))}{\sum_{j=1}^2 (1+\omega(\varpi_j))}(-\log(1+\Psi_{\varpi_2}))^\pi\right)^{\frac{1}{\pi}}}\right) + i\left(-1 + \left(\mathbb{e}^{-\left(\frac{(1+\omega(\varpi_2))}{\sum_{j=1}^2 (1+\omega(\varpi_j))}(-\log(1+\Phi_{\varpi_2}))^\pi\right)^{\frac{1}{\pi}}}\right)\right)
\end{array}
\right)
$$

Thus, by using the above information, we have

$$
BCFLAAPOG\left(\varpi_1^{BC}, \varpi_2^{BC}\right) = \left(\varpi_1^{BC}\right)^{\overline{\frac{(1+\omega(\varpi_1))}{\sum_{j=1}^2 (1+\omega(\varpi_j))}}} \otimes \left(\varpi_2^{BC}\right)^{\overline{\frac{(1+\omega(\varpi_2))}{\sum_{j=1}^2 (1+\omega(\varpi_j))}}}
$$

$$
= \left(
\begin{array}{c}
\mathcal{L}_\partial\left(\mathbb{e}^{-\left(\frac{(1+\omega(\varpi_1))}{\sum_{j=1}^2 (1+\omega(\varpi_j))}\left(-\log\left(\frac{\mathfrak{l}_1}{\partial}\right)\right)^\pi\right)^{\frac{1}{\pi}}}\right), \\[2em]
\left(\left(\mathbb{e}^{-\left(\frac{(1+\omega(\varpi_1))}{\sum_{j=1}^2 (1+\omega(\varpi_j))}(-\log(\Xi_{\varpi_1}))^\pi\right)^{\frac{1}{\pi}}}\right) + i\left(\mathbb{e}^{-\left(\frac{(1+\omega(\varpi_1))}{\sum_{j=1}^2 (1+\omega(\varpi_j))}(-\log(\Omega_{\varpi_1}))^\pi\right)^{\frac{1}{\pi}}}\right)\right), \\[2em]
-1 + \left(\mathbb{e}^{-\left(\frac{(1+\omega(\varpi_1))}{\sum_{j=1}^2 (1+\omega(\varpi_j))}(-\log(1+\Psi_{\varpi_1}))^\pi\right)^{\frac{1}{\pi}}}\right) + i\left(-1 + \left(\mathbb{e}^{-\left(\frac{(1+\omega(\varpi_1))}{\sum_{j=1}^2 (1+\omega(\varpi_j))}(-\log(1+\Phi_{\varpi_1}))^\pi\right)^{\frac{1}{\pi}}}\right)\right)
\end{array}
\right)
$$
$$
\otimes \left(
\begin{array}{c}
\mathcal{L}_\partial\left(\mathbb{e}^{-\left(\frac{(1+\omega(\varpi_2))}{\sum_{j=1}^2 (1+\omega(\varpi_j))}\left(-\log\left(\frac{\mathfrak{l}_2}{\partial}\right)\right)^\pi\right)^{\frac{1}{\pi}}}\right), \\[2em]
\left(\left(\mathbb{e}^{-\left(\frac{(1+\omega(\varpi_2))}{\sum_{j=1}^2 (1+\omega(\varpi_j))}(-\log(\Xi_{\varpi_2}))^\pi\right)^{\frac{1}{\pi}}}\right) + i\left(\mathbb{e}^{-\left(\frac{(1+\omega(\varpi_2))}{\sum_{j=1}^2 (1+\omega(\varpi_j))}(-\log(\Omega_{\varpi_2}))^\pi\right)^{\frac{1}{\pi}}}\right)\right), \\[2em]
-1 + \left(\mathbb{e}^{-\left(\frac{(1+\omega(\varpi_2))}{\sum_{j=1}^2 (1+\omega(\varpi_j))}(-\log(1+\Psi_{\varpi_2}))^\pi\right)^{\frac{1}{\pi}}}\right) + i\left(-1 + \left(\mathbb{e}^{-\left(\frac{(1+\omega(\varpi_2))}{\sum_{j=1}^2 (1+\omega(\varpi_j))}(-\log(1+\Phi_{\varpi_2}))^\pi\right)^{\frac{1}{\pi}}}\right)\right)
\end{array}
\right)
$$

$$= \begin{pmatrix} \mathcal{L} \\ \partial \left( \mathbb{e}^{-\left( \sum_{j=1}^{2} \frac{(1+\omega(\varpi_{j}))}{\sum_{j=1}^{n}(1+\omega(\varpi_{j}))} \left( -\log\left( \frac{l_{j}}{\partial} \right) \right)^{\pi} \right)^{\frac{1}{\pi}}} \right), \\ \left( \mathbb{e}^{-\left( \sum_{j=1}^{2} \frac{(1+\omega(\varpi_{j}))}{\sum_{j=1}^{n}(1+\omega(\varpi_{j}))} \left( -\log(\Xi_{\varpi_{j}}) \right)^{\pi} \right)^{\frac{1}{\pi}}} \right) + i \left( \mathbb{e}^{-\left( \sum_{j=1}^{2} \frac{(1+\omega(\varpi_{j}))}{\sum_{j=1}^{n}(1+\omega(\varpi_{j}))} \left( -\log(\Omega_{\varpi_{j}}) \right)^{\pi} \right)^{\frac{1}{\pi}}} \right), \\ -1 + \left( \mathbb{e}^{-\left( \sum_{j=1}^{2} \frac{(1+\omega(\varpi_{j}))}{\sum_{j=1}^{n}(1+\omega(\varpi_{j}))} \left( -\log(1+\Psi_{\varpi_{j}}) \right)^{\pi} \right)^{\frac{1}{\pi}}} \right) + i \left( -1 + \left( \mathbb{e}^{-\left( \sum_{j=1}^{2} \frac{(1+\omega(\varpi_{j}))}{\sum_{j=1}^{n}(1+\omega(\varpi_{j}))} \left( -\log(1+\Phi_{\varpi_{j}}) \right)^{\pi} \right)^{\frac{1}{\pi}}} \right) \right) \end{pmatrix}$$

Hence, the proposed model is held for $n = 2$. Further, we assume that the proposed model also holds for $n = n'$, such as

$BCFLAAPOG(\varpi_{1}, \varpi_{2}, \ldots, \varpi_{n'})$

$$= \begin{pmatrix} \mathcal{L} \\ \partial \left( \mathbb{e}^{-\left( \sum_{j=1}^{n'} \frac{(1+\omega(\varpi_{j}))}{\sum_{j=1}^{n'}(1+\omega(\varpi_{j}))} \left( -\log\left( \frac{l_{j}}{\partial} \right) \right)^{\pi} \right)^{\frac{1}{\pi}}} \right), \\ \left( \mathbb{e}^{-\left( \sum_{j=1}^{n'} \frac{(1+\omega(\varpi_{j}))}{\sum_{j=1}^{n'}(1+\omega(\varpi_{j}))} \left( -\log(\Xi_{\varpi_{j}}) \right)^{\pi} \right)^{\frac{1}{\pi}}} \right) + i \left( \mathbb{e}^{-\left( \sum_{j=1}^{n'} \frac{(1+\omega(\varpi_{j}))}{\sum_{j=1}^{n'}(1+\omega(\varpi_{j}))} \left( -\log(\Omega_{\varpi_{j}}) \right)^{\pi} \right)^{\frac{1}{\pi}}} \right), \\ -1 + \left( \mathbb{e}^{-\left( \sum_{j=1}^{n'} \frac{(1+\omega(\varpi_{j}))}{\sum_{j=1}^{n'}(1+\omega(\varpi_{j}))} \left( -\log(1+\Psi_{\varpi_{j}}) \right)^{\pi} \right)^{\frac{1}{\pi}}} \right) + i \left( -1 + \left( \mathbb{e}^{-\left( \sum_{j=1}^{n'} \frac{(1+\omega(\varpi_{j}))}{\sum_{j=1}^{n'}(1+\omega(\varpi_{j}))} \left( -\log(1+\Phi_{\varpi_{j}}) \right)^{\pi} \right)^{\frac{1}{\pi}}} \right) \right) \end{pmatrix}$$

Then, we proved that the proposed model also holds for $n = n'+1$, such as

$$BCFLAAPOG(\varpi_{1}^{BC}, \varpi_{2}^{BC}, \ldots, \varpi_{n'+1}^{BC}) = BCFLAAPOG(\varpi_{1}, \varpi_{2}, \ldots, \varpi_{n'+1})$$

$$= \otimes_{j=1}^{n'} (\varpi_{j}^{BC})^{\frac{(1+\omega(\varpi_{j}))}{\sum_{j=1}^{n'}(1+\omega(\varpi_{j}))}} \otimes (\varpi_{n'+1}^{BC})^{\frac{(1+\omega(\varpi_{n'+1}))}{\sum_{j=1}^{n'+1}(1+\omega(\varpi_{j}))}}$$

$$
= \left( \begin{array}{c}
\mathcal{L}_{\partial}\left( \mathbb{e}^{-\left(\sum_{j=1}^{n'} \frac{(1+\omega(\varpi_j))}{\sum_{j=1}^{n'}(1+\omega(\varpi_j))}\left(-\log\left(\frac{l_j}{\partial}\right)\right)^{\pi}\right)^{\frac{1}{\pi}}} \right), \\[2em]
\left( \left( \mathbb{e}^{-\left(\sum_{j=1}^{n'} \frac{(1+\omega(\varpi_j))}{\sum_{j=1}^{n'}(1+\omega(\varpi_j))}\left(-\log(\Xi_{\varpi_j})\right)^{\pi}\right)^{\frac{1}{\pi}}} \right) + i\left( \mathbb{e}^{-\left(\sum_{j=1}^{n'} \frac{(1+\omega(\varpi_j))}{\sum_{j=1}^{n'}(1+\omega(\varpi_j))}\left(-\log(\Omega_{\varpi_j})\right)^{\pi}\right)^{\frac{1}{\pi}}} \right), \\[2em]
-1 + \left( \mathbb{e}^{-\left(\sum_{j=1}^{n'} \frac{(1+\omega(\varpi_j))}{\sum_{j=1}^{n'}(1+\omega(\varpi_j))}\left(-\log(1+\Psi_{\varpi_j})\right)^{\pi}\right)^{\frac{1}{\pi}}} \right) + i\left( -1 + \left( \mathbb{e}^{-\left(\sum_{j=1}^{n'} \frac{(1+\omega(\varpi_j))}{\sum_{j=1}^{n'}(1+\omega(\varpi_j))}\left(-\log(1+\Phi_{\varpi_j})\right)^{\pi}\right)^{\frac{1}{\pi}}} \right) \right) \right)
\end{array} \right)
$$

$$
\otimes \left(\varpi_{n'+1}^{BC}\right)^{\frac{(1+\omega(\varpi_{n'+1}))}{\sum_{j=1}^{n'+1}(1+\omega(\varpi_j))}}
$$

$$
= \left( \begin{array}{c}
\mathcal{L}_{\partial}\left( \mathbb{e}^{-\left(\sum_{j=1}^{n'} \frac{(1+\omega(\varpi_j))}{\sum_{j=1}^{n'}(1+\omega(\varpi_j))}\left(-\log\left(\frac{l_j}{\partial}\right)\right)^{\pi}\right)^{\frac{1}{\pi}}} \right), \\[2em]
\left( \left( \mathbb{e}^{-\left(\sum_{j=1}^{n'} \frac{(1+\omega(\varpi_j))}{\sum_{j=1}^{n'}(1+\omega(\varpi_j))}\left(-\log(\Xi_{\varpi_j})\right)^{\pi}\right)^{\frac{1}{\pi}}} \right) + i\left( \mathbb{e}^{-\left(\sum_{j=1}^{n'} \frac{(1+\omega(\varpi_j))}{\sum_{j=1}^{n'}(1+\omega(\varpi_j))}\left(-\log(\Omega_{\varpi_j})\right)^{\pi}\right)^{\frac{1}{\pi}}} \right), \\[2em]
-1 + \left( \mathbb{e}^{-\left(\sum_{j=1}^{n'} \frac{(1+\omega(\varpi_j))}{\sum_{j=1}^{n'}(1+\omega(\varpi_j))}\left(-\log(1+\Psi_{\varpi_j})\right)^{\pi}\right)^{\frac{1}{\pi}}} \right) + i\left( -1 + \left( \mathbb{e}^{-\left(\sum_{j=1}^{n'} \frac{(1+\omega(\varpi_j))}{\sum_{j=1}^{n'}(1+\omega(\varpi_j))}\left(-\log(1+\Phi_{\varpi_j})\right)^{\pi}\right)^{\frac{1}{\pi}}} \right) \right) \right)
\end{array} \right)
$$

$$
\oplus \left( \begin{array}{c}
\mathcal{L}_{\partial}\left( \mathbb{e}^{-\left(\frac{(1+\omega(\varpi_{n'+1}))}{\sum_{j=1}^{n'+1}(1+\omega(\varpi_j))}\left(-\log\left(\frac{l_{n'+1}}{\partial}\right)\right)^{\pi}\right)^{\frac{1}{\pi}}} \right), \\[2em]
\left( \left( \mathbb{e}^{-\left(\frac{(1+\omega(\varpi_{n'+1}))}{\sum_{j=1}^{n'+1}(1+\omega(\varpi_j))}\left(-\log(\Xi_{\varpi_{n'+1}})\right)^{\pi}\right)^{\frac{1}{\pi}}} \right) + i\left( \mathbb{e}^{-\left(\frac{(1+\omega(\varpi_{n'+1}))}{\sum_{j=1}^{n'+1}(1+\omega(\varpi_j))}\left(-\log(\Omega_{\varpi_{n'+1}})\right)^{\pi}\right)^{\frac{1}{\pi}}} \right), \\[2em]
-1 + \left( \mathbb{e}^{-\left(\frac{(1+\omega(\varpi_{n'+1}))}{\sum_{j=1}^{n'+1}(1+\omega(\varpi_j))}\left(-\log(1+\Psi_{\varpi_{n'+1}})\right)^{\pi}\right)^{\frac{1}{\pi}}} \right) + i\left( -1 + \left( \mathbb{e}^{-\left(\frac{(1+\omega(\varpi_{n'+1}))}{\sum_{j=1}^{n'+1}(1+\omega(\varpi_j))}\left(-\log(1+\Phi_{\varpi_{n'+1}})\right)^{\pi}\right)^{\frac{1}{\pi}}} \right) \right) \right)
\end{array} \right)
$$

$$
= \left(
\begin{array}{c}
\mathcal{L}_{\partial} \left( e^{-\left( \sum_{j=1}^{n'+1} \frac{(1+\omega(\varpi_j))}{\sum_{j=1}^{n'+1}(1+\omega(\varpi_j))} \left(-\log\left(\frac{l_j}{\partial}\right)\right)^\pi \right)^{\frac{1}{\pi}}} \right), \\[2em]
\left( e^{-\left( \sum_{j=1}^{n'+1} \frac{(1+\omega(\varpi_j))}{\sum_{j=1}^{n'+1}(1+\omega(\varpi_j))} \left(-\log(\Xi_{\varpi_j})\right)^\pi \right)^{\frac{1}{\pi}}} \right) + i\left( e^{-\left( \sum_{j=1}^{n'+1} \frac{(1+\omega(\varpi_j))}{\sum_{j=1}^{n'+1}(1+\omega(\varpi_j))} \left(-\log(\Omega_{\varpi_j})\right)^\pi \right)^{\frac{1}{\pi}}} \right), \\[2em]
-1 + \left( e^{-\left( \sum_{j=1}^{n'+1} \frac{(1+\omega(\varpi_j))}{\sum_{j=1}^{n'+1}(1+\omega(\varpi_j))} \left(-\log(1+\Psi_{\varpi_j})\right)^\pi \right)^{\frac{1}{\pi}}} \right) + i\left( -1 + \left( e^{-\left( \sum_{j=1}^{n'+1} \frac{(1+\omega(\varpi_j))}{\sum_{j=1}^{n'+1}(1+\omega(\varpi_j))} \left(-\log(1+\Phi_{\varpi_j})\right)^\pi \right)^{\frac{1}{\pi}}} \right) \right)
\end{array}
\right)
$$

Hence, the proposed theory holds for $n = n'+1$, it means that the proposed theory holds for non-negative integers. Moreover, we simplify some basic or major properties for the above operators, called idempotency, monotonicity, and boundedness.

**Property 3:** Let $\varpi_j^{BC} = (\mathcal{L}_{l_j}, \Xi_{\varpi_j} + i\Omega_{\varpi_j}, \Psi_{\varpi_j} + i\Phi_{\varpi_j}), j = 1, 2, \ldots, n$ be the collection of BCFLNs. Then,

1. Idempotency: When $\varpi_j = \varpi, j = 1, 2, \ldots, n$, thus

$$
BCFLAAPOG(\varpi_1, \varpi_2, \ldots, \varpi_n) = \varpi
$$

2. Monotonicity: When $\varpi_j \leq \varpi_j^{@}, j = 1, 2, \ldots, n$, thus

$$
BCFLAAPOG(\varpi_1, \varpi_2, \ldots, \varpi_n) \leq BCFLAAPOG(\varpi_1^{@}, \varpi_2^{@}, \ldots, \varpi_n^{@})
$$

3. Boundedness: When $\varpi_j^{-} = \min_j(\varpi_j)$ and $\varpi_j^{+} = \max_j(\varpi_j)$, thus

$$
\varpi_j^{-} \leq BCFLAAPOG(\varpi_1, \varpi_2, \ldots, \varpi_n) \leq \varpi_j^{+}
$$

**Proof:** The proof of this Property is similar to the proof of Property 1.

**Definition 8:** For any finite family of BCFLNs
$\varpi_j^{BC} = (\mathcal{L}_{l_j}, \Xi_{\varpi_j} + i\Omega_{\varpi_j}, \Psi_{\varpi_j} + i\Phi_{\varpi_j}), j = 1, 2, \ldots, n$, the model of the BCFLAAPOWG operator is described and illustrated below:

$$
BCFLAAPOWG : Z^n \to Z
$$

by

$$BCFLAAPOWG(\varpi_1, \varpi_2, \ldots, \varpi_n)$$

$$= \varpi_1^{\frac{\Psi_1(1+\omega(\varpi_1))}{\sum_{j=1}^{n} \Psi_j(1+\omega(\varpi_j))}} \otimes \varpi_2^{\frac{\Psi_2(1+\omega(\varpi_2))}{\sum_{j=1}^{n} \Psi_j(1+\omega(\varpi_j))}} \otimes \varpi_3^{\frac{\Psi_3(1+\omega(\varpi_3))}{\sum_{j=1}^{n} \Psi_j(1+\omega(\varpi_j))}} \otimes \ldots \otimes \varpi_n^{\frac{\Psi_n(1+\omega(\varpi_n))}{\sum_{j=1}^{n} \Psi_j(1+\omega(\varpi_j))}}$$

$$= \otimes_{j=1}^{n} (\varpi_j)^{\frac{\Psi_j(1+\omega(\varpi_j))}{\sum_{j=1}^{n} \Psi_j(1+\omega(\varpi_j))}}$$

Where $\omega(\varpi_j) = \sum_{j \neq k=1}^{n} SP(\varpi_j, \varpi_k)$ and $SP(\varpi_j, \varpi_k) = 1 - DS(\varpi_j, \varpi_k)$, with some conditions:

1. $SP(\varpi_j, \varpi_k) \in [0, 1]$.

2. $SP(\varpi_j, \varpi_k) = SP(\varpi_k, \varpi_j)$.

3. If $SP(\varpi_j, \varpi_k) < SP(\varpi_l, \varpi_m)$ then $DS(\varpi_j, \varpi_k) \geq DS(\varpi_l, \varpi_m)$.

Further,

$$DS\left(\varpi_j, \varpi_k\right) = \frac{1}{2}\left(\frac{|\mathfrak{l}_j - \mathfrak{l}_k|}{\partial} + \frac{1}{4}\left(|\Xi_{\varpi_j} - \Xi_{\varpi_k}| + |\Omega_{\varpi_j} - \Omega_{\varpi_k}| + |\Psi_{\varpi_j} - \Psi_{\varpi_k}| + |\Phi_{\varpi_j} - \Phi_{\varpi_k}|\right)\right)$$

Where the weight vector is represented by $\Psi_j \in [0,1]$ with a condition that is $\sum_{j=1}^{n} \Psi_j = 1$. The term $Z^n$ contained the collection of BCFLNs. Further, by using the information in Def. (4), we aim to calculate the aggregated values of the information in Def. (8).

**Theorem 4:** Let $\varpi_j^{BC} = (\mathcal{L}_{\mathfrak{l}_j}, \Xi_{\varpi_j} + i\Omega_{\varpi_j}, \Psi_{\varpi_j} + i\Phi_{\varpi_j}), j = 1, 2, \ldots, n$ be the collection of BCFLNs. Then, by using the information in Def. (4) and Def. (8), we proved that their aggregated value is also a BCFLN, such as

$BCFLAAPOWG(\varpi_1, \varpi_2, \ldots, \varpi_n)$

$$= \left( \mathcal{L}_{\partial\left(\mathbb{e}^{-\left(\sum_{j=1}^{n} \frac{\Psi_j(1+\omega(\varpi_j))}{\sum_{j=1}^{n} \Psi_j(1+\omega(\varpi_j))}\left(-\log\left(\frac{\mathfrak{l}_j}{\partial}\right)\right)^{\pi}\right)^{\frac{1}{\pi}}}\right)}, \right.$$

$$\left( \mathbb{e}^{-\left(\sum_{j=1}^{n} \frac{\Psi_j(1+\omega(\varpi_j))}{\sum_{j=1}^{n} \Psi_j(1+\omega(\varpi_j))}\left(-\log(\Xi_{\varpi_j})\right)^{\pi}\right)^{\frac{1}{\pi}}} \right) + i\left( \mathbb{e}^{-\left(\sum_{j=1}^{n} \frac{\Psi_j(1+\omega(\varpi_j))}{\sum_{j=1}^{n} \Psi_j(1+\omega(\varpi_j))}\left(-\log(\Omega_{\varpi_j})\right)^{\pi}\right)^{\frac{1}{\pi}}} \right),$$

$$\left. -1 + \left( \mathbb{e}^{-\left(\sum_{j=1}^{n} \frac{\Psi_j(1+\omega(\varpi_j))}{\sum_{j=1}^{n} \Psi_j(1+\omega(\varpi_j))}\left(-\log(1+\Psi_{\varpi_j})\right)^{\pi}\right)^{\frac{1}{\pi}}} \right) + i\left( -1 + \left( \mathbb{e}^{-\left(\sum_{j=1}^{n} \frac{\Psi_j(1+\omega(\varpi_j))}{\sum_{j=1}^{n} \Psi_j(1+\omega(\varpi_j))}\left(-\log(1+\Phi_{\varpi_j})\right)^{\pi}\right)^{\frac{1}{\pi}}} \right) \right) \right)$$

**Proof:** The proof of this Theorem is similar to the proof of the Theorem 3. Moreover, we simplify some basic or major properties for the above operators, called idempotency, monotonicity, and boundedness.

**Property 4:** Let $\varpi_j^{BC} = (\mathcal{L}_{\mathfrak{l}_j}, \Xi_{\varpi_j} + i\Omega_{\varpi_j}, \Psi_{\varpi_j} + i\Phi_{\varpi_j}), j = 1, 2, \ldots, n$ be the collection of BCFLNs. Then,

1. Idempotency: When $\varpi_j = \varpi, j = 1, 2, \ldots, n$, thus

$$BCFLAAPOWG(\varpi_1, \varpi_2, \ldots, \varpi_n) = \varpi$$

2. Monotonicity: When $\varpi_j \leq \varpi_j^{@}, j = 1, 2, \ldots, n$, thus

$$BCFLAAPOWG(\varpi_1, \varpi_2, \ldots, \varpi_n) \leq BCFLAAPOWG(\varpi_1^{@}, \varpi_2^{@}, \ldots, \varpi_n^{@})$$

3. Boundedness: When $\varpi_j^- = \min_j(\varpi_j)$ and $\varpi_j^+ = \max_j(\varpi_j)$, thus

$$\varpi_j^- \leq BCFLAAPOWG(\varpi_1, \varpi_2, \ldots, \varpi_n) \leq \varpi_j^+$$

**Proof:** The proof of this Property is similar to the proof of Property 1. Further, we simplify the information in Theorem 4 by using some numerical examples. Consider $\varpi_1 = (\mathcal{L}_{0.2087}, 0.0945 + i(0.126), -0.732 + i(-0.791)), \varpi_2 = (\mathcal{L}_{0.2314}, 0.0974 + i(0.1294), -0.739 + i(-0.797)),$ $\varpi_3 = (\mathcal{L}_{0.2545}, 0.1004 + i(0.1329), -0.745 + i(-0.801)), \varpi_4 = (\mathcal{L}_{0.2779}, 0.1034 + i(0.1364),$ $-0.752 + i(-0.806)),$ and $\varpi_5 = (\mathcal{L}_{0.3018}, 0.1065 + i(0.14), -0.758 + i(-0.811)),$ with $\Psi = (0.3, 0.2, 0.1, 0.3, 0.1)^T$ represents the weight vector and $\pi = 2$, thus

$$
\begin{aligned}
SP(\varpi_1, \varpi_2) &= 1 - DS(\varpi_1, \varpi_2) \\
&= 1 \\
&\quad - \frac{1}{2}\left(\frac{|\mathfrak{l}_1 - \mathfrak{l}_2|}{\partial} + \frac{1}{4}\left(|\Xi_{\varpi_1} - \Xi_{\varpi_2}| + |\Omega_{\varpi_1} - \Omega_{\varpi_2}| + |\Psi_{\varpi_1} - \Psi_{\varpi_2}| + |\Phi_{\varpi_1} - \Phi_{\varpi_2}|\right)\right) \\
&= 0.9958, SP(\varpi_1, \varpi_3) = 0.9916, SP(\varpi_1, \varpi_4) = 0.9874, SP(\varpi_1, \varpi_5) \\
&= 0.9833, SP(\varpi_2, \varpi_3) = 0.9958, SP(\varpi_2, \varpi_4) = 0.9916, SP(\varpi_2, \varpi_5) \\
&= 0.9874, SP(\varpi_3, \varpi_4) = 0.9958, SP(\varpi_3, \varpi_5) = 0.9916, SP(\varpi_4, \varpi_5) = 0.9958.
\end{aligned}
$$

Further,

$$
\begin{aligned}
\omega(\varpi_1) &= \sum_{j \neq k=1}^{5} SP(\varpi_1, \varpi_k) = SP(\varpi_1, \varpi_2) + SP(\varpi_1, \varpi_3) + SP(\varpi_1, \varpi_4) + SP(\varpi_1, \varpi_5) \\
&= 0.9958 + 0.9916 + 0.9874 + 0.9833 = 3.9581, \omega(\varpi_2) = 3.9707, \omega(\varpi_3) \\
&= 3.9749, \omega(\varpi_4) = 3.9707, \omega(\varpi_5) = 3.9581
\end{aligned}
$$

$$
\begin{aligned}
&\frac{\Psi_1(1 + \omega(\varpi_1))}{\sum_{j=1}^{5} \Psi_j(1 + \omega(\varpi_j))} = \frac{0.3*(1 + 3.9707)}{24.833} = 0.2995, \frac{\Psi_2(1 + \omega(\varpi_2))}{\sum_{j=1}^{5} \Psi_j(1 + \omega(\varpi_j))} \\
&= 0.2002, \frac{\Psi_3(1 + \omega(\varpi_3))}{\sum_{j=1}^{5} \Psi_j(1 + \omega(\varpi_j))} 0.1002, \frac{\Psi_4(1 + \omega(\varpi_4))}{\sum_{j=1}^{5} \Psi_j(1 + \omega(\varpi_j))} 0.3003, \frac{\Psi_5(1 + \omega(\varpi_5))}{\sum_{j=1}^{5} \Psi_j(1 + \omega(\varpi_j))} 0.0998
\end{aligned}
$$

Thus,

$$BCFLAAPOWG(\varpi_1, \varpi_2, \varpi_3, \varpi_4, \varpi_5)$$

$$= \begin{pmatrix} \mathcal{L}_{\partial\left(\mathbb{e}^{-\left(\sum_{j=1}^n \frac{\Psi_j(1+\omega(\varpi_j))}{\sum_{j=1}^n \Psi_j(1+\omega(\varpi_j))}\left(-\log\left(\frac{l_j}{\partial}\right)\right)^\pi\right)^{\frac{1}{\pi}}}\right)}, \\ \left(\mathbb{e}^{-\left(\sum_{j=1}^n \frac{\Psi_j(1+\omega(\varpi_j))}{\sum_{j=1}^n \Psi_j(1+\omega(\varpi_j))}(-\log(\Xi_{\varpi_j}))^\pi\right)^{\frac{1}{\pi}}}\right) + i\left(\mathbb{e}^{-\left(\sum_{j=1}^n \frac{\Psi_j(1+\omega(\varpi_j))}{\sum_{j=1}^n \Psi_j(1+\omega(\varpi_j))}(-\log(\Omega_{\varpi_j}))^\pi\right)^{\frac{1}{\pi}}}\right), \\ -1 + \left(\mathbb{e}^{-\left(\sum_{j=1}^n \frac{\Psi_j(1+\omega(\varpi_j))}{\sum_{j=1}^n \Psi_j(1+\omega(\varpi_j))}(-\log(1+\Psi_{\varpi_j}))^\pi\right)^{\frac{1}{\pi}}}\right) + i\left(-1 + \left(\mathbb{e}^{-\left(\sum_{j=1}^n \frac{\Psi_j(1+\omega(\varpi_j))}{\sum_{j=1}^n \Psi_j(1+\omega(\varpi_j))}(-\log(1+\Phi_{\varpi_j}))^\pi\right)^{\frac{1}{\pi}}}\right)\right) \end{pmatrix}$$

$$= (\mathcal{L}_{1.4962}, 0.2669 + i(0.4147), -0.446 + i(-0.503)).$$

## 4. WASPAS method for proposed operators

In this section, we compute the technique WASPAS based on the initiated operators by using the information of BCFL values. In the investigation of the WASPAS method, we used the BCFLAAPOA operator, BCFLAAPOWA operator, BCFLAAPOG operator, and BCFLAA-POWG operator to enhance the worth of the proposed theory. The major steps of the WAS-PAS technique are listed below:

Step 1: Defining a matrix consisting of bipolar complex fuzzy linguistic variables

$$\mathbb{M} = \begin{bmatrix} \varpi_{11} & \varpi_{12} & \varpi_{13} & \cdots & \varpi_{1n} \\ \varpi_{21} & \varpi_{22} & \varpi_{23} & \cdots & \varpi_{2n} \\ \varpi_{31} & \varpi_{32} & \varpi_{33} & \cdots & \varpi_{3n} \\ \cdots & \cdots & \cdots & \cdots & \cdots \\ \varpi_{m1} & \varpi_{m2} & \varpi_{m3} & \cdots & \varpi_{mn} \end{bmatrix}$$

Where, $\varpi_{ij}^{BC} = (\mathcal{L}_{l_{ij}}, \Xi_{\varpi_{ij}} + i\Omega_{\varpi_{ij}}, \Psi_{\varpi_{ij}} + i\Phi_{\varpi_{ij}}), i, j = 1, 2, \ldots, m, n$. The geometrical representation of the WASPAS technique is illustrated in the shape of Fig 2.

$$\overline{\overline{\mathcal{L}}}_{l_{ij}} = \mathcal{L}_{\frac{l_{ij}}{\max(l_{ij})}}, \text{if } \max_i(l_{ij}) \text{ is preferable}$$

$$\overline{\overline{\Xi}}_{\varpi_{ij}} = \frac{\Xi_{\varpi_{ij}}}{\max(\Xi_{\varpi_{ij}})}, = \Omega_{\varpi_{ij}} = \frac{\Omega_{\varpi_{ij}}}{\max(\Omega_{\varpi_{ij}})}, \text{if } \max(\Xi_{\varpi_{ij}}), \max(\Omega_{\varpi_{ij}}) \text{ is preferable}$$

$$\overline{\overline{\Psi}}_{\varpi_{ij}} = \frac{-|\Psi_{\varpi_{ij}}|}{\max(|\Psi_{\varpi_{ij}}|)}, = \Phi_{\varpi_{ij}} = \frac{-|\Phi_{\varpi_{ij}}|}{\max(|\Phi_{\varpi_{ij}}|)}, \text{if } \max(|\Psi_{\varpi_{ij}}|), \max(|\Phi_{\varpi_{ij}}|) \text{ is preferable}$$

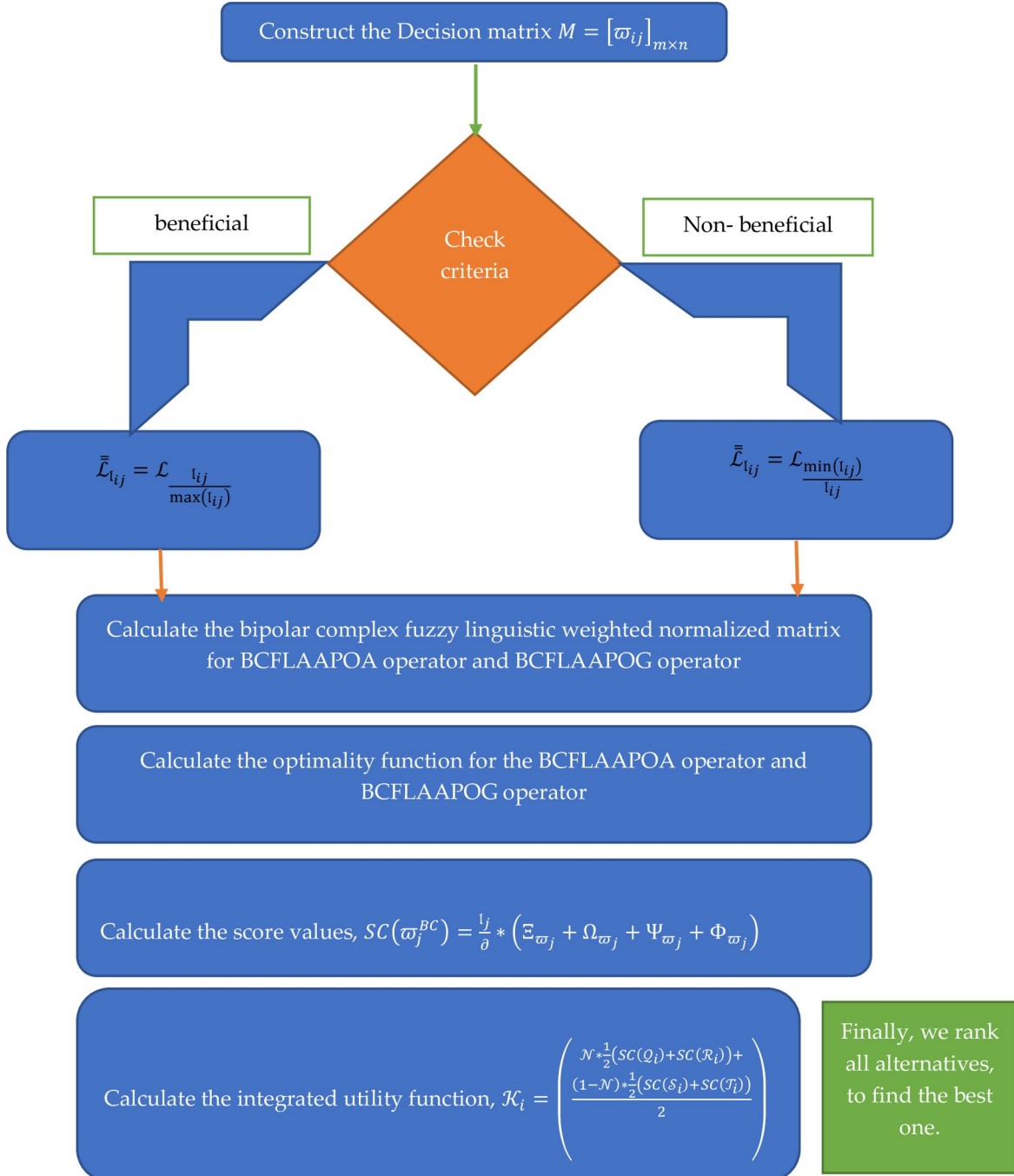

**Fig 2. Geometrical representation of the WASPAS technique.**

And

$$\overline{\overline{\mathcal{L}}}_{\mathfrak{l}_{ij}} = \mathcal{L}_{\frac{\min(\mathfrak{l}_{ij})}{\mathfrak{l}_{ij}}}, \text{ if } \min_i(\mathfrak{l}_{ij}) \text{ is preferable}$$

$$\overline{\overline{\Xi}}_{\varpi_{ij}} = \frac{\min(\Xi_{\varpi_{ij}})}{\Xi_{\varpi_{ij}}}, = \Omega_{\varpi_{ij}} = \frac{\min(\Omega_{\varpi_{ij}})}{\Omega_{\varpi_{ij}}}, \text{if } \min(\Xi_{\varpi_{ij}}), \min(\Omega_{\varpi_{ij}}) \text{ is preferable}$$

$$\overline{\overline{\Psi}}_{\varpi_{ij}} = -\frac{\min(|\Psi_{\varpi_{ij}}|)}{|\Psi_{\varpi_{ij}}|}, = \Phi_{\varpi_{ij}} = -\frac{\min(|\Phi_{\varpi_{ij}}|)}{|\Phi_{\varpi_{ij}}|}, \text{if } \min(|\Psi_{\varpi_{ij}}|), \min(|\Phi_{\varpi_{ij}}|) \text{ is preferable}$$

Step 3: Calculate the bipolar complex fuzzy linguistic weighted normalized matrix for BCFLAAPOA operator and BCFLAAPOG operator, such as

$$\mathcal{L}_{=\mathfrak{l}_{ij}} = \mathcal{L}_{\partial * \left(1 - \mathbb{e}^{-\left(\Psi_j\left(-\log\left(1-\frac{\mathfrak{l}_{ij}}{\partial}\right)\right)^\pi\right)^{\frac{1}{\pi}}}\right)}, \overline{\overline{\Xi}}_{\varpi_{ij}} = 1 - \left(\mathbb{e}^{-(\Psi_j(-\log(1-\Xi_{\varpi_{ij}}))^\pi)^{\frac{1}{\pi}}}\right), \overline{\overline{\Xi}}_{\varpi_{ij}}$$

$$= 1 - \left(\mathbb{e}^{-(\Psi_j(-\log(1-\Omega_{\varpi_{ij}}))^\pi)^{\frac{1}{\pi}}}\right), \overline{\overline{\Psi}}_{\varpi_{ij}} = -\left(\mathbb{e}^{-(\Psi_j(-\log(|\Psi_{\varpi_{ij}}|))^\pi)^{\frac{1}{\pi}}}\right), = \Phi_{\varpi_{ij}}$$

$$= -\left(\mathbb{e}^{-(\Psi_j(-\log(|\Phi_{\varpi_{ij}}|))^\pi)^{\frac{1}{\pi}}}\right)$$

And

$$\mathcal{L}_{=\mathfrak{l}_{ij}} = \mathcal{L}_{\partial * \left(\mathbb{e}^{-\left(\Psi_j\left(-\log\left(\frac{\mathfrak{l}_{ij}}{\partial}\right)\right)^\pi\right)^{\frac{1}{\pi}}}\right)}, \overline{\overline{\Xi}}_{\varpi_{ij}} = \left(\mathbb{e}^{-(\Psi_j(-\log(\Xi_{\varpi_{ij}}))^\pi)^{\frac{1}{\pi}}}\right), \overline{\overline{\Xi}}_{\varpi_{ij}} = \left(\mathbb{e}^{-(\Psi_j(-\log(\Omega_{\varpi_{ij}}))^\pi)^{\frac{1}{\pi}}}\right), \overline{\overline{\Psi}}_{\varpi_{ij}}$$

$$= -1 + \left(\mathbb{e}^{-(\Psi_j(-\log(1+\Psi_{\varpi_{ij}}))^\pi)^{\frac{1}{\pi}}}\right), = \Phi_{\varpi_{ij}} = -1 + \left(\mathbb{e}^{-(\Psi_j(-\log(1+\Phi_{\varpi_{ij}}))^\pi)^{\frac{1}{\pi}}}\right)$$

Step 4: Calculate the optimality function for the BCFLAAPOA operator and BCFLAAPOG operator, such as

$$\mathcal{Q}_i = BCFLAAPOA(\varpi_{i1}^{BC}, \varpi_{i2}^{BC}, \ldots, \varpi_{im}^{BC})$$

$$\mathcal{R}_i = BCFLAAPOWA(\varpi_{i1}^{BC}, \varpi_{i2}^{BC}, \ldots, \varpi_{im}^{BC})$$

$$\mathcal{S}_i = BCFLAAPOG(\varpi_{i1}^{BC}, \varpi_{i2}^{BC}, \ldots, \varpi_{im}^{BC})$$

$$\mathcal{T}_i = BCFLAAPOWG(\varpi_{i1}^{BC}, \varpi_{i2}^{BC}, \ldots, \varpi_{im}^{BC})$$

Step 5: Calculate the score values of all aggregate functions, such as

$$SC\left(\varpi_j^{BC}\right) = \frac{\mathfrak{l}_j}{\partial} * \left(\Xi_{\varpi_j} + \Omega_{\varpi_j} + \Psi_{\varpi_j} + \Phi_{\varpi_j}\right)$$

Step 6: Calculate the integrated utility function of the WASPAS method, such as

$$\mathcal{K}_i = \frac{\mathcal{N} * \frac{1}{2}(SC(\mathcal{Q}_i) + SC(\mathcal{R}_i)) + (1-\mathcal{N}) * \frac{1}{2}(SC(\mathcal{S}_i) + SC(\mathcal{T}_i))}{2}$$

Step 7: Rank all the alternatives and examine the best one.

## 5. Coupling in geographic information systems for BCFL-WASPAS method

In the context of geography information systems, the technique model of "coupling" is used for the investigation of the degree of integration among different systems or components with a geography information system. Effective and reliable coupling in geography information systems improves the system's functionality, flexibility, usability, and validity, leading to more efficient spatial information management and systems. This section aims to evaluate the problem of coupling in geographic information systems based on the WASPAS technique for BCFL variables. The WASPAS technique is computed based on initiated techniques, called BCFLAAPOA operator, BCFLAAPOWA operator, BCFLAAPOG operator, and BCFLAA-POWG operator. Coupling in geographic information systems is a very beneficial technique for evaluating or analyzing the relationship between different components within a geographic information system. In this application, we used five levels of geographic information systems and with the help of the WASPAS technique, we found the best one, such as

1. Data Coupling "$\varpi_1^{BC}$": The data coupling is used for how spatial and attribute information are linked within the geographic information systems.

2. Data Capture and Input "$\varpi_2^{BC}$": Data capture and input I geographic information systems are used to processes by which spatial and criteria information are arranged and entered into geographic information systems.

3. Modeling and Simulation "$\varpi_3^{BC}$": Modeling and simulation are very important techniques used in geographic information systems and many other fields to understand, predict, and analyze complicated systems and procedures.

4. Functional Coupling "$\varpi_4^{BC}$": Functional coupling in geographic information systems is used for the integration and interaction among different functional modules or components within a geographic information system.

5. Spatial Coupling "$\varpi_5^{BC}$": Spatial coupling in geographic information systems is used for the way different spatial datasets or layers interact and relate to each other within a geographic information system.

Further, for the selection of the above alternative we will use the following attributes, such as growth analysis, social impact, political impact, environmental impact, and internet resources with weight vectors $(0.3, 0.2, 0.1, 0.3, 0.1)^T$. The major steps of the WASPAS technique are listed below:

Step 1: Defining a matrix consisting of bipolar complex fuzzy linguistic variables, see Table 1.

Where, $\varpi_{ij}^{BC} = (\mathcal{L}_{I_{ij}}, \Xi_{\varpi_{ij}} + i\Omega_{\varpi_{ij}}, \Psi_{\varpi_{ij}} + i\Phi_{\varpi_{ij}}), i, j = 1, 2, \ldots, m, n.$

Step 2: Normalize the bipolar fuzzy linguistic decision matrix, see Table 2.

Step 3: Calculate the bipolar complex fuzzy linguistic weighted normalized matrix for the BCFLAAPOA operator and BCFLAAPOG operator, see Table 3.

Step 4: Calculate the optimality function for the BCFLAAPOA operator, BCFLAAPOWA operator, BCFLAAPOG operator, and BCFLAAPOWG operator, see Table 4.

Step 5: Calculate the score values of all aggregate functions, see Table 5.

Step 6: Calculate the integrated utility function of the WASPAS method, such as

$$\mathcal{K}_1 = -0.01942, \mathcal{K}_2 = 0.00212, \mathcal{K}_3 = 0.01252, \mathcal{K}_4 = -0.01259, \mathcal{K}_5 = -0.07755$$

**Table 1. BCFL decision matrix.**

| | $\varpi_1^{AT}$ | $\varpi_2^{AT}$ | $\varpi_3^{AT}$ | $\varpi_4^{AT}$ | $\varpi_5^{AT}$ |
|---|---|---|---|---|---|
| $\varpi_1^{BC}$ | $\begin{pmatrix} \mathcal{L}_1, \\ 0.4 + i(0.5), \\ -0.2 + i(-0.3) \end{pmatrix}$ | $\begin{pmatrix} \mathcal{L}_{1.1}, \\ 0.41 + i(0.51), \\ -0.21 + i(-0.31) \end{pmatrix}$ | $\begin{pmatrix} \mathcal{L}_{1.2}, \\ 0.42 + i(0.52), \\ -0.22 + i(-0.32) \end{pmatrix}$ | $\begin{pmatrix} \mathcal{L}_{1.3}, \\ 0.43 + i(0.53), \\ -0.23 + i(-0.33) \end{pmatrix}$ | $\begin{pmatrix} \mathcal{L}_{1.4}, \\ 0.44 + i(0.54), \\ -0.24 + i(-0.34) \end{pmatrix}$ |
| $\varpi_2^{BC}$ | $\begin{pmatrix} \mathcal{L}_2, \\ 0.7 + i(0.8), \\ -0.4 + i(-0.2) \end{pmatrix}$ | $\begin{pmatrix} \mathcal{L}_{2.1}, \\ 0.71 + i(0.81), \\ -0.41 + i(-0.21) \end{pmatrix}$ | $\begin{pmatrix} \mathcal{L}_{2.2}, \\ 0.72 + i(0.82), \\ -0.42 + i(-0.22) \end{pmatrix}$ | $\begin{pmatrix} \mathcal{L}_{2.3}, \\ 0.73 + i(0.83), \\ -0.43 + i(-0.23) \end{pmatrix}$ | $\begin{pmatrix} \mathcal{L}_{2.4}, \\ 0.74 + i(0.84), \\ -0.44 + i(-0.24) \end{pmatrix}$ |
| $\varpi_3^{BC}$ | $\begin{pmatrix} \mathcal{L}_3, \\ 0.8 + i(0.9), \\ -0.3 + i(-0.4) \end{pmatrix}$ | $\begin{pmatrix} \mathcal{L}_{3.1}, \\ 0.81 + i(0.91), \\ -0.31 + i(-0.41) \end{pmatrix}$ | $\begin{pmatrix} \mathcal{L}_{3.2}, \\ 0.82 + i(0.92), \\ -0.32 + i(-0.42) \end{pmatrix}$ | $\begin{pmatrix} \mathcal{L}_{3.3}, \\ 0.83 + i(0.93), \\ -0.33 + i(-0.43) \end{pmatrix}$ | $\begin{pmatrix} \mathcal{L}_{3.4}, \\ 0.84 + i(0.94), \\ -0.34 + i(-0.44) \end{pmatrix}$ |
| $\varpi_4^{BC}$ | $\begin{pmatrix} \mathcal{L}_4, \\ 0.5 + i(0.6), \\ -0.2 + i(-0.1) \end{pmatrix}$ | $\begin{pmatrix} \mathcal{L}_{4.1}, \\ 0.51 + i(0.61), \\ -0.21 + i(-0.11) \end{pmatrix}$ | $\begin{pmatrix} \mathcal{L}_{4.2}, \\ 0.52 + i(0.62), \\ -0.22 + i(-0.12) \end{pmatrix}$ | $\begin{pmatrix} \mathcal{L}_{4.3}, \\ 0.53 + i(0.63), \\ -0.23 + i(-0.13) \end{pmatrix}$ | $\begin{pmatrix} \mathcal{L}_{4.4}, \\ 0.54 + i(0.64), \\ -0.24 + i(-0.14) \end{pmatrix}$ |
| $\varpi_5^{BC}$ | $\begin{pmatrix} \mathcal{L}_5, \\ 0.6 + i(0.7), \\ -0.5 + i(-0.4) \end{pmatrix}$ | $\begin{pmatrix} \mathcal{L}_{5.1}, \\ 0.61 + i(0.71), \\ -0.51 + i(-0.41) \end{pmatrix}$ | $\begin{pmatrix} \mathcal{L}_{5.2}, \\ 0.62 + i(0.72), \\ -0.52 + i(-0.42) \end{pmatrix}$ | $\begin{pmatrix} \mathcal{L}_{5.3}, \\ 0.63 + i(0.73), \\ -0.53 + i(-0.43) \end{pmatrix}$ | $\begin{pmatrix} \mathcal{L}_{5.4}, \\ 0.64 + i(0.74), \\ -0.54 + i(-0.44) \end{pmatrix}$ |

**Table 2. Normalized BCFL decision matrix.**

| | $\varpi_1^{AT}$ | $\varpi_2^{AT}$ | $\varpi_3^{AT}$ | $\varpi_4^{AT}$ | $\varpi_5^{AT}$ |
|---|---|---|---|---|---|
| $\varpi_1^{BC}$ | $\begin{pmatrix} \mathcal{L}_{07142.}, \\ 0.9090+ \\ i(0.9259), \\ -0.8333+ \\ i(-0.8823) \end{pmatrix}$ | $\begin{pmatrix} \mathcal{L}_{0.7857}, \\ 0.9318+ \\ i(0.9444), \\ -0.875+ \\ i(-0.9117) \end{pmatrix}$ | $\begin{pmatrix} \mathcal{L}_{0.8571}, \\ 0.9545+ \\ i(0.9629), \\ -0.9166+ \\ i(-0.9411) \end{pmatrix}$ | $\begin{pmatrix} \mathcal{L}_{0.9285}, \\ 0.9772+ \\ i(0.9814), \\ -0.9583+ \\ i(-0.9705) \end{pmatrix}$ | $\begin{pmatrix} \mathcal{L}_1, \\ 1+ \\ i(1), \\ -1+ \\ i(-1) \end{pmatrix}$ |
| $\varpi_2^{BC}$ | $\begin{pmatrix} \mathcal{L}_{0.8333}, \\ 0.9459+ \\ i(0.9523), \\ -0.9090+ \\ i(-0.8333) \end{pmatrix}$ | $\begin{pmatrix} \mathcal{L}_{0.875}, \\ 0.9594+ \\ i(0.9642), \\ -0.9218+ \\ i(-0.875) \end{pmatrix}$ | $\begin{pmatrix} \mathcal{L}_{0.9166}, \\ 0.9729+ \\ i(0.9761), \\ -0.9545+ \\ i(-0.9166) \end{pmatrix}$ | $\begin{pmatrix} \mathcal{L}_{0.9583}, \\ 0.9864+ \\ i(0.9881), \\ -0.9772+ \\ i(-0.9583) \end{pmatrix}$ | $\begin{pmatrix} \mathcal{L}_1, \\ 1+ \\ i(1), \\ -1+ \\ i(-1) \end{pmatrix}$ |
| $\varpi_3^{BC}$ | $\begin{pmatrix} \mathcal{L}_{0.8823}, \\ 0.9523+ \\ i(0.9574), \\ -0.8823+ \\ i(-0.9090) \end{pmatrix}$ | $\begin{pmatrix} \mathcal{L}_{0.9117}, \\ 0.9642+ \\ i(0.9680), \\ -0.9117+ \\ i(-0.9318) \end{pmatrix}$ | $\begin{pmatrix} \mathcal{L}_{0.9411}, \\ 0.9761+ \\ i(0.9787), \\ -0.9411+ \\ i(-0.9545) \end{pmatrix}$ | $\begin{pmatrix} \mathcal{L}_{0.9705}, \\ 0.9881+ \\ i(0.9893), \\ -0.9705+ \\ i(-0.9772) \end{pmatrix}$ | $\begin{pmatrix} \mathcal{L}_1, \\ 1+ \\ i(1), \\ -1+ \\ i(-1) \end{pmatrix}$ |
| $\varpi_4^{BC}$ | $\begin{pmatrix} \mathcal{L}_{0.9090}, \\ 0.9259+ \\ i(0.9375), \\ -0.8333+ \\ i(-0.7142) \end{pmatrix}$ | $\begin{pmatrix} \mathcal{L}_{0.9318}, \\ 0.9444+ \\ i(0.9531), \\ -0.875+ \\ i(-0.7857) \end{pmatrix}$ | $\begin{pmatrix} \mathcal{L}_{0.9545}, \\ 0.9629+ \\ i(0.9687), \\ -0.9166+ \\ i(-0.8571) \end{pmatrix}$ | $\begin{pmatrix} \mathcal{L}_{0.9772}, \\ 0.9814+ \\ i(0.9843), \\ -0.9583+ \\ i(-0.9285) \end{pmatrix}$ | $\begin{pmatrix} \mathcal{L}_1, \\ 1+ \\ i(1), \\ -1+ \\ i(-1) \end{pmatrix}$ |
| $\varpi_5^{BC}$ | $\begin{pmatrix} \mathcal{L}_{0.9259}, \\ 0.9375+ \\ i(0.9459), \\ -0.9259+ \\ i(-0.9090) \end{pmatrix}$ | $\begin{pmatrix} \mathcal{L}_{0.9444}, \\ 0.9531+ \\ i(0.9594), \\ -0.9444+ \\ i(-0.9318) \end{pmatrix}$ | $\begin{pmatrix} \mathcal{L}_{0.9629}, \\ 0.9687+ \\ i(0.9729), \\ -0.9629+ \\ i(-0.9545) \end{pmatrix}$ | $\begin{pmatrix} \mathcal{L}_{0.9814}, \\ 0.9843+ \\ i(0.9864), \\ -0.9814+ \\ i(-0.9772) \end{pmatrix}$ | $\begin{pmatrix} \mathcal{L}_1, \\ 1+ \\ i(1), \\ -1+ \\ i(-1) \end{pmatrix}$ |

**Table 3. Weighted normalized BCFL decision matrix.**

| | $\varpi_1^{AT}$ | $\varpi_2^{AT}$ | $\varpi_3^{AT}$ | $\varpi_4^{AT}$ | $\varpi_5^{AT}$ |
|---|---|---|---|---|---|
| $\varpi_1^{BC}$ | $\begin{pmatrix} \mathcal{L}_{0.2087}, \\ 0.0944+ \\ i(0.1259), \\ -0.7315+ \\ i(-0.7914) \end{pmatrix}$ | $\begin{pmatrix} \mathcal{L}_{0.2314}, \\ 0.0974+ \\ i(0.1293), \\ -0.7385+ \\ i(-0.7965) \end{pmatrix}$ | $\begin{pmatrix} \mathcal{L}_{0.2544}, \\ 0.1003+ \\ i(0.1328), \\ -0.7452+ \\ i(-0.8014) \end{pmatrix}$ | $\begin{pmatrix} \mathcal{L}_{0.2779}, \\ 0.1034+ \\ i(0.1364), \\ -0.7516+ \\ i(-0.8062) \end{pmatrix}$ | $\begin{pmatrix} \mathcal{L}_{0.3017}, \\ 0.1065+ \\ i(0.14), \\ -0.7579+ \\ i(-0.8109) \end{pmatrix}$ |
| $\varpi_2^{BC}$ | $\begin{pmatrix} \mathcal{L}_{0.4543}, \\ 0.2085+ \\ i(0.2684), \\ -0.8369+ \\ i(-0.7315) \end{pmatrix}$ | $\begin{pmatrix} \mathcal{L}_{0.4815}, \\ 0.2137+ \\ i(0.2757), \\ -0.841+ \\ i(-0.7385) \end{pmatrix}$ | $\begin{pmatrix} \mathcal{L}_{0.5093}, \\ 0.2190+ \\ i(0.2832), \\ -0.8449+ \\ i(-0.7452) \end{pmatrix}$ | $\begin{pmatrix} \mathcal{L}_{0.5377}, \\ 0.2245+ \\ i(0.2911), \\ -0.8488+ \\ i(-0.7516) \end{pmatrix}$ | $\begin{pmatrix} \mathcal{L}_{0.5667}, \\ 0.2302+ \\ i(0.2994), \\ -0.8526+ \\ i(-0.7579) \end{pmatrix}$ |
| $\varpi_3^{BC}$ | $\begin{pmatrix} \mathcal{L}_{0.7557}, \\ 0.2684+ \\ i(0.3605), \\ -0.7914+ \\ i(-0.8369) \end{pmatrix}$ | $\begin{pmatrix} \mathcal{L}_{0.7901}, \\ 0.2757+ \\ i(0.3735), \\ -0.7965+ \\ i(-0.841) \end{pmatrix}$ | $\begin{pmatrix} \mathcal{L}_{0.8255}, \\ 0.2832+ \\ i(0.3877), \\ -0.8014+ \\ i(-0.8449) \end{pmatrix}$ | $\begin{pmatrix} \mathcal{L}_{0.8619}, \\ 0.2911+ \\ i(0.4033), \\ -0.8062+ \\ i(-0.8488) \end{pmatrix}$ | $\begin{pmatrix} \mathcal{L}_{0.8994}, \\ 0.2994+ \\ i(0.4209), \\ -0.8109+ \\ i(-0.8526) \end{pmatrix}$ |
| $\varpi_4^{BC}$ | $\begin{pmatrix} \mathcal{L}_{1.1528}, \\ 0.1259+ \\ i(0.1630), \\ -0.7315+ \\ i(-0.6394) \end{pmatrix}$ | $\begin{pmatrix} \mathcal{L}_{1.2009}, \\ 0.1293+ \\ i(0.1671), \\ -0.7385+ \\ i(-0.6513) \end{pmatrix}$ | $\begin{pmatrix} \mathcal{L}_{1.2510}, \\ 0.1328+ \\ i(0.1713), \\ -0.7452+ \\ i(-0.6624) \end{pmatrix}$ | $\begin{pmatrix} \mathcal{L}_{1.3034}, \\ 0.1364+ \\ i(0.1756), \\ -0.7516+ \\ i(-0.6728) \end{pmatrix}$ | $\begin{pmatrix} \mathcal{L}_{1.3584}, \\ 0.14+ \\ i(0.1799), \\ -0.7579+ \\ i(-0.6825) \end{pmatrix}$ |
| $\varpi_5^{BC}$ | $\begin{pmatrix} \mathcal{L}_{1.7634}, \\ 0.1630+ \\ i(0.2085), \\ -0.8740+ \\ i(-0.8369) \end{pmatrix}$ | $\begin{pmatrix} \mathcal{L}_{1.8492}, \\ 0.1671+ \\ i(0.2137), \\ -0.8774+ \\ i(-0.841) \end{pmatrix}$ | $\begin{pmatrix} \mathcal{L}_{1.9431}, \\ 0.1713+ \\ i(0.2190), \\ -0.8807+ \\ i(-0.8449) \end{pmatrix}$ | $\begin{pmatrix} \mathcal{L}_{2.0469}, \\ 0.1756+ \\ i(0.2245), \\ -0.8839+ \\ i(-0.8488) \end{pmatrix}$ | $\begin{pmatrix} \mathcal{L}_{2.1635}, \\ 0.1799+ \\ i(0.2302), \\ -0.8872+ \\ i(-0.8526) \end{pmatrix}$ |

Step 7: Rank all the alternatives and examine the best one, such as

$$\mathcal{K}_3 > \mathcal{K}_2 > \mathcal{K}_4 > \mathcal{K}_1 > \mathcal{K}_5$$

The most valuable and dominant decision is Modeling and Simulation "$\varpi_3^{BC}$" between the above fives. Further, we calculate the ranking values by using the proposed operators without following the WASPAS technique, then the ranking result is given in the shape of Table 6 (see Table 5).

The most valuable and dominant decision is Modeling and Simulation "$\varpi_3^{BC}$" according to the theory of the BCFLAAPOG operator and BCFLAAPOWG operator, but in the consideration of the BCFLAAPOA operator and BCFLAAPOWA operator, the best decision is Data Coupling "$\varpi_1^{BC}$". Additionally, we described the significant of the parameters by using the technique of proposed BCFLAAPOWG operator are described in Table 7.

For different values of parameter, the proposed BCFLAAPOWG operator is given the same best optimal which states the stability of the initiated operators. Further, we will evaluate the supremacy of the proposed theory by comparing our results with some existing techniques.

**Table 4. Aggregated BCFL matrix.**

| | BCFLAAPOA | BCFLAAPOWA | BCFLAAPOG | BCFLAAPOWG |
|---|---|---|---|---|
| $\varpi_1^{BC}$ | $\begin{pmatrix} \mathcal{L}_{0.1130}, \\ 0.0449+ \\ i(0.0601), \\ -0.8798+ \\ i(-0.9082) \end{pmatrix}$ | $\begin{pmatrix} \mathcal{L}_{0.1099}, \\ 0.0445+ \\ i(0.0596), \\ -0.8788+ \\ i(-0.9075) \end{pmatrix}$ | $\begin{pmatrix} \mathcal{L}_{1.5145}, \\ 0.3683+ \\ i(0.4160), \\ -0.4478+ \\ i(-0.5045) \end{pmatrix}$ | $\begin{pmatrix} \mathcal{L}_{1.4962}, \\ 0.3669+ \\ i(0.4146), \\ -0.4459+ \\ i(-0.5029) \end{pmatrix}$ |
| $\varpi_2^{BC}$ | $\begin{pmatrix} \mathcal{L}_{0.2278}, \\ 0.1019+ \\ i(0.1350), \\ -0.9293+ \\ i(-0.8798) \end{pmatrix}$ | $\begin{pmatrix} \mathcal{L}_{0.2240}, \\ 0.1011+ \\ i(0.1338), \\ -0.9287+ \\ i(-0.8788) \end{pmatrix}$ | $\begin{pmatrix} \mathcal{L}_{2.0529}, \\ 0.5170+ \\ i(0.5781), \\ -0.5550+ \\ i(-0.4478) \end{pmatrix}$ | $\begin{pmatrix} \mathcal{L}_{2.0381}, \\ 0.5154+ \\ i(0.5761), \\ -0.5535+ \\ i(-0.4459) \end{pmatrix}$ |
| $\varpi_3^{BC}$ | $\begin{pmatrix} \mathcal{L}_{0.3749}, \\ 0.1350+ \\ i(0.1934), \\ -0.9082+ \\ i(-0.9293) \end{pmatrix}$ | $\begin{pmatrix} \mathcal{L}_{0.3699}, \\ 0.1338+ \\ i(0.1908), \\ -0.9075+ \\ i(-0.9287) \end{pmatrix}$ | $\begin{pmatrix} 2.5336, \\ 0.5781+ \\ i(0.6629), \\ -0.5045+ \\ i(-0.5550) \end{pmatrix}$ | $\begin{pmatrix} \mathcal{L}_{2.5193}, \\ 0.5761+ \\ i(0.6596), \\ -0.5029+ \\ i(-0.5535) \end{pmatrix}$ |
| $\varpi_4^{BC}$ | $\begin{pmatrix} \mathcal{L}_{0.582}, \\ 0.0601+ \\ i(0.0784), \\ -0.8798+ \\ i(-0.8355) \end{pmatrix}$ | $\begin{pmatrix} \mathcal{L}_{0.5744}, \\ 0.0596+ \\ i(0.0778), \\ -0.8788+ \\ i(-0.8337) \end{pmatrix}$ | $\begin{pmatrix} \mathcal{L}_{3.0358}, \\ 0.4160+ \\ i(0.4647), \\ -0.4478+ \\ i(-0.3759) \end{pmatrix}$ | $\begin{pmatrix} \mathcal{L}_{3.0197}, \\ 0.4146+ \\ i(0.4632), \\ -0.4459+ \\ i(-0.3733) \end{pmatrix}$ |
| $\varpi_5^{BC}$ | $\begin{pmatrix} \mathcal{L}_{0.9479}, \\ 0.0784+ \\ i(0.1019), \\ -0.9462+ \\ i(-0.9293) \end{pmatrix}$ | $\begin{pmatrix} \mathcal{L}_{0.9318}, \\ 0.0778+ \\ i(0.1011), \\ -0.9462+ \\ i(-0.9293) \end{pmatrix}$ | $\begin{pmatrix} \mathcal{L}_{3.6774}, \\ 0.4646+ \\ i(0.5170), \\ -0.6029+ \\ i(-0.5550) \end{pmatrix}$ | $\begin{pmatrix} \mathcal{L}_{3.6534}, \\ 0.4632+ \\ i(0.5154), \\ -0.6015+ \\ i(-0.5535) \end{pmatrix}$ |

**Table 5. Score values of the aggregated values.**

| | $\varpi_1^{AT}$ | $\varpi_2^{AT}$ | $\varpi_3^{AT}$ | $\varpi_4^{AT}$ |
|---|---|---|---|---|
| $\varpi_1^{BC}$ | −0.03171 | −0.03084 | −0.0424 | −0.04174 |
| $\varpi_2^{BC}$ | −0.0597 | −0.5872 | 0.03161 | 0.03125 |
| $\varpi_3^{BC}$ | −0.09431 | −0.0932 | 0.07665 | 0.07525 |
| $\varpi_4^{BC}$ | −0.15295 | −0.15079 | 0.02882 | 0.02944 |
| $\varpi_5^{BC}$ | −0.26783 | −0.26334 | −0.10801 | −0.10747 |

**Table 6. Ranking values of the aggregated values.**

| Methods | Ranking values | Best decision |
|---|---|---|
| BCFLAAPOA Operator | $\mathcal{K}_1 > \mathcal{K}_2 > \mathcal{K}_3 > \mathcal{K}_4 > \mathcal{K}_5$ | $\mathcal{K}_1$ |
| BCFLAAPOWA Operator | $\mathcal{K}_1 > \mathcal{K}_3 > \mathcal{K}_4 > \mathcal{K}_5 > \mathcal{K}_2$ | $\mathcal{K}_1$ |
| BCFLAAPOG Operator | $\mathcal{K}_3 > \mathcal{K}_2 > \mathcal{K}_4 > \mathcal{K}_1 > \mathcal{K}_5$ | $\mathcal{K}_3$ |
| BCFLAAPOWG Operator | $\mathcal{K}_3 > \mathcal{K}_2 > \mathcal{K}_4 > \mathcal{K}_1 > \mathcal{K}_5$ | $\mathcal{K}_3$ |

**Table 7. Stability of the parameter using the BCFLAAPOWG operator.**

| Parameter | Score values | Ranking values |
|---|---|---|
| $\pi = 2$ | -0.0194,0.0021,0.0125,-0.0126,-0.0775 | $\mathcal{K}_3 > \mathcal{K}_2 > \mathcal{K}_4 > \mathcal{K}_1 > \mathcal{K}_5$ |
| $\pi = 4$ | 0.0077,0.048,0.0708,0.0491,-0.0211 | $\mathcal{K}_3 > \mathcal{K}_4 > \mathcal{K}_2 > \mathcal{K}_1 > \mathcal{K}_5$ |
| $\pi = 6$ | 0.0195,0.0674,0.0949,0.0749,0.0032 | $\mathcal{K}_3 > \mathcal{K}_4 > \mathcal{K}_2 > \mathcal{K}_1 > \mathcal{K}_5$ |
| $\pi = 8$ | 0.0258,0.0777,0.1077,0.0866,0.0161 | $\mathcal{K}_3 > \mathcal{K}_4 > \mathcal{K}_2 > \mathcal{K}_1 > \mathcal{K}_5$ |
| $\pi = 10$ | 0.0297,0.0841,0.1154,0.0969,0.024 | $\mathcal{K}_3 > \mathcal{K}_4 > \mathcal{K}_2 > \mathcal{K}_1 > \mathcal{K}_5$ |

## 6. Comparative analysis

This section aims to compare the proposed ranking values with the obtained ranking values to enhance or show the supremacy and validity of the proposed theory. For comparison, we will use the information in Table 1, and then have the following existing techniques, such as Mahmood et al. [25] proposed aggregation operators for BCFLSs. Further, Yager [26] evaluated power operators for crisp values. Mardani et al. [29] presented the WASPAS theory for FSs and Jaleel [30] exposed it for BCFSs. Bi et al. [31] initiated the arithmetic operators for CFSs and Hu et al. [32] derived the power operators for CFSs. Jana et al. [33] exposed the Dombi operators for BFSs. Mahmood et al. [34] evaluated the aggregation operators for BCFSs. Further, Mahmood et al. [36] presented the Aczel-Alsina operators for BCFSs. Thus, based on the data in Table 1, the comparative analysis is listed in Table 8.

The most valuable and dominant decision is Modeling and Simulation "$\varpi_3^{BC}$" according to the theory of the BCFLAAPOG operator, BCFLAAPOWG operator, Mahmood et al. [25], and WASPAS method, but in the consideration of the BCFLAAPOA operator and BCFLAA-POWA operator, the best decision is Data Coupling "$\varpi_1^{BC}$". Further, the existing techniques and methods failed to evaluate the data in Table 1, because these are the special cases of the proposed operators. The major advantages of the presented techniques are listed below:

1. Maximum, minimum, Drastic, Algebraic, power, Aczel-Alsina aggregation operators, and WASPAS method for fuzzy information.

2. Maximum, minimum, Drastic, Algebraic, power, Aczel-Alsina aggregation operators, and WASPAS method for bipolar fuzzy information.

**Table 8. Comparative analysis between proposed and prevailing information.**

| Methods | Ranking values | Best decision |
|---|---|---|
| Mahmood et al. [25] | $\mathcal{K}_3 > \mathcal{K}_2 > \mathcal{K}_4 > \mathcal{K}_1 > \mathcal{K}_5$ | $\mathcal{K}_3$ |
| Yager [26] | Failed to evaluate the data in Table 1. | no |
| Mardani et al. [29] | Failed to evaluate the data in Table 1. | no |
| Jaleel [30] | Failed to evaluate the data in Table 1. | no |
| Bi et al. [31] | Failed to evaluate the data in Table 1. | no |
| Hu et al. [32] | Failed to evaluate the data in Table 1. | no |
| Jana et al. [33] | Failed to evaluate the data in Table 1. | no |
| Mahmood et al. [34] | Failed to evaluate the data in Table 1. | no |
| Mahmood et al. [36] | Failed to evaluate the data in Table 1. | no |
| WASPAS Method | $\mathcal{K}_3 > \mathcal{K}_2 > \mathcal{K}_4 > \mathcal{K}_1 > \mathcal{K}_5$ | $\mathcal{K}_3$ |
| BCFLAAPOA Operator | $\mathcal{K}_1 > \mathcal{K}_2 > \mathcal{K}_3 > \mathcal{K}_4 > \mathcal{K}_5$ | $\mathcal{K}_1$ |
| BCFLAAPOWA Operator | $\mathcal{K}_1 > \mathcal{K}_3 > \mathcal{K}_4 > \mathcal{K}_5 > \mathcal{K}_2$ | $\mathcal{K}_1$ |
| BCFLAAPOG Operator | $\mathcal{K}_3 > \mathcal{K}_2 > \mathcal{K}_4 > \mathcal{K}_1 > \mathcal{K}_5$ | $\mathcal{K}_3$ |
| BCFLAAPOWG Operator | $\mathcal{K}_3 > \mathcal{K}_2 > \mathcal{K}_4 > \mathcal{K}_1 > \mathcal{K}_5$ | $\mathcal{K}_3$ |

3. Maximum, minimum, Drastic, Algebraic, power, Aczel-Alsina aggregation operators, and WASPAS method for linguistic term information.

4. Maximum, minimum, Drastic, Algebraic, power, Aczel-Alsina aggregation operators, and WASPAS method for complex fuzzy information.

5. Maximum, minimum, Drastic, Algebraic, power, Aczel-Alsina aggregation operators, and WASPAS method for bipolar fuzzy information.

6. Maximum, minimum, Drastic, Algebraic, power, Aczel-Alsina aggregation operators, and WASPAS method for bipolar complex fuzzy information.

7. Maximum, minimum, Drastic, Algebraic, power, Aczel-Alsina aggregation operators, and WASPAS method for fuzzy linguistic term information.

8. Maximum, minimum, Drastic, Algebraic, power, Aczel-Alsina aggregation operators, and WASPAS method for bipolar fuzzy linguistic term information.

9. Maximum, minimum, Drastic, Algebraic, power, Aczel-Alsina aggregation operators, and WASPAS method for complex fuzzy linguistic term information.

10. Maximum, minimum, Drastic, Algebraic, power, Aczel-Alsina aggregation operators, and WASPAS method for bipolar complex fuzzy linguistic term information.

Form the above analysis, we clear that the proposed theory is very powerful and dominant because of its features, but the existing techniques are contained a lot of problems. The limitations of the existing techniques are briefly discussed below:

1. Mahmood et al. [25] proposed aggregation operators for BCFLSs, where the ranking values of the proposed theory of Mahmood et al. [25] follow, such as $\mathcal{K}_3 > \mathcal{K}_2 > \mathcal{K}_4 > \mathcal{K}_1 > \mathcal{K}_5$, but the proposed theory is the modified version of the existing techniques of Mahmood et al. [25].

2. Yager [26] evaluated power operators for crisp values has failed to cope with it because of unlimited complications and problems, where the technique of Yager [26] is a special part of the proposed theory.

3. Mardani et al. [29] presented that the WASPAS theory for FSs has failed to cope with it because of unlimited complications and problems, where the technique of Mardani et al. [29] is the special part of the proposed theory.

4. Jaleel [30] exposed it for BCFSs. Bi et al. [31] initiated the arithmetic operators for CFSs and Hu et al. [32] derived the power operators for CFSs. Jana et al. [33] exposed the Dombi operators for BFSs. Mahmood et al. [34] evaluated the aggregation operators for BCFSs. Further, Mahmood et al. [36] presented the Aczel-Alsina operators for BCFSs. These techniques are the modified version of the proposed theory.

Hence, the existing techniques have failed to cope with it, but the proposed theory is more general and more modified than the existing information to cope with it.

## 7. Conclusion

The model of BCFL information is very strong and reliable to cope with vague and uncertain information in genuine-life problems. The techniques of Aczel-Alsina aggregation operators, power aggregation operators, and the WASPAS technique have various advantages. Based on their features, we have described the following ideas, such as

1. We diagnosed the model of Aczel-Alsina operational laws for BCFL variables.

2. We evaluated the BCFLAAPOA operator, BCFLAAPOWA operator, BCFLAAPOG operator, and BCFLAAPOWG operator.

3. We presented some fundamental properties for the above operators.

4. We computed the WASPAS method based on initiated operators.

5. We demonstrated the procedure of the MADM technique based on initiated operators for computing the best technique for addressing geographic information systems.

6. We compared the ranking values of initiated techniques with the ranking values of the existing techniques based on illustrated examples to enhance the worth of the proposed theory.

## 7.1. Limitations of the proposed model

The model of the BCFL technique is very proficient because of the positive membership function, negative membership function, and linguistic variable, but in the presence of complex information the technique of the BCFL set is not working, for instance, when we provide the positive and negative information in the form of membership and non-membership, then the technique of BCFL set has been failed. For this, we need to provide the model of bipolar complex intuitionistic fuzzy linguistic sets and their extensions.

## 7.2. Future directions

In the future, we will discuss the Hamacher operators, Einstein operators, Dombi operators, and Frank operators based on bipolar complex fuzzy linguistic sets and also try to propose some new techniques, called TOPSIS method, MARCOS methods, VIKOR method, AHP method, and CODAS method to enhance the worth of the proposed theory.

## Acknowledgments

This paper is supported by the NRPU-HEC Pakistan Project Number 14662 and the joint project PSF(PSF-NSFC/JSEP/ENG/AJKUKAJK/01)-NSFC (12211540710).

## Author Contributions

**Data curation:** Khizar Hayat.

**Methodology:** Zeeshan Ali, Khizar Hayat.

**Software:** Zeeshan Ali.

**Supervision:** Khizar Hayat.

**Validation:** Zeeshan Ali.

**Visualization:** Khizar Hayat.

**Writing – original draft:** Zeeshan Ali, Khizar Hayat.

**Writing – review & editing:** Zeeshan Ali, Khizar Hayat, Dragan Pamucar.

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
