## [Decision Letter · Decision Letter 0]

8 Jul 2024

PONE-D-24-24446Analysis of Coupling in Geographic Information Systems Based on WASPAS Method for Bipolar Complex Fuzzy Linguistic Aczel-Alsina Power Aggregation OperatorsPLOS ONE

Dear Dr. Pamucar,

Thank you for submitting your manuscript to PLOS ONE. After careful consideration, we feel that it has merit but does not fully meet PLOS ONE’s publication criteria as it currently stands. Therefore, we invite you to submit a revised version of the manuscript that addresses the points raised during the review process.

We look forward to receiving your revised manuscript.

Kind regards,

Muhammet Gul, Ph.D.

Academic Editor

PLOS ONE

Journal Requirements:

**Comments from PLOS Editorial Office: **

We note that one or more reviewers has recommended that you cite specific previously published works. As always, we recommend that you please review and evaluate the requested works to determine whether they are relevant and should be cited. It is not a requirement to cite these works. We appreciate your attention to this request.

Reviewers' comments:

Reviewer's Responses to Questions

**Comments to the Author**

1. Is the manuscript technically sound, and do the data support the conclusions?

Reviewer #1: Yes

Reviewer #2: Yes

2. Has the statistical analysis been performed appropriately and rigorously? 

Reviewer #1: Yes

Reviewer #2: N/A

3. Have the authors made all data underlying the findings in their manuscript fully available?

Reviewer #1: Yes

Reviewer #2: No

4. Is the manuscript presented in an intelligible fashion and written in standard English?

Reviewer #1: Yes

Reviewer #2: Yes

5. Review Comments to the Author

Reviewer #1: I have read the manuscript very carefully, the manuscript is very interesting and results

are corrected. After accepting the paper, I have some minor revision, such as:

1) Avoid abbreviations in abstract.

2) In the introduction section, add advantages, clearly present research gap.

3) Extended the literature review section by presenting relevant methods, e.g. (parsimonious spherical fuzzy AHP and the integrated IMF SWARA and Fuzzy Bonferroni operator)

4) present results in figures

5) In Section 2, please include some descriptive analysis for each definition.

6) The authors suggested checking the grammar carefully.

7) The order of the citation in your manuscript is not the proper way, please correct it.

8) In the application, add relatedgeometrical diagrams.

Reviewer #2: This manuscript presents an approach applying Aczel-Alsina aggregation operators to bipolar complex fuzzy linguistic sets for analyzing coupling in geographic information systems. However, the paper requires major revisions before it is suitable for publication. I have the following concerns:

- The abstract does not clearly summarize the key contributions and objectives of the paper.

- The integration of Aczel-Alsina operators with bipolar complex fuzzy linguistic sets appears novel and could provide valuable tools for decision making under uncertainty. However, the novelty claim is not fully justified. - a more extensive literature review is needed to clearly situate this work relative to previous studies applying fuzzy approaches to GIS coupling analysis.

-The literature review appears to be incomplete and lacks critical analysis of existing work in the field. The authors should: Expand the review to include more recent publications (within the last 3-5 years). Provide a more in-depth analysis of how their work relates to and builds upon previous research. Clearly identify the research gap their study aims to fill. Add the recent work such as MADM models[10.3389/frai.2024.1347626, 10.22105/jfea.2023.426042.1331, 10.61356/j.mawa.2024.4241, 10.56578/josa020203, 10.1007/s00500-023-09459-0, 10.22105/jarie.2022.339957.1467, 10.31181/sa2120246, 10.22105/jfea.2023.425520.1326, 10.61356/j.mawa.2024.26761]; Bipolar/ linguistic sets[10.1007/s40314-023-02254-5, ]; aggregation operators [10.56578/josa020103, 10.22105/jfea.2023.422582.1318, 10.1007/s40314-021-01651-y, 10.3934/math.2023577, 10.1038/s41598-023-37497-z, 10.22105/bdcv.2021.142090 ]; various fuzzy set extensions[10.1111/exsy.12542, 10.22105/jfea.2024.424633.1328, 10.3934/math.2022954, 10.22105/bdcv.2023.192676, 10.3233/JIFS-210655, 10.22111/ijfs.2024.45118.7968, ].

- The methodology could be explained more clearly, especially the definitions of the various operators which rely on multiple mathematical terms defined earlier. More intuitive explanations may help non-expert readers. Provide a more detailed explanation of the BCFLAAPOA and BCFLAAPOG operators. Justify the choice of these specific operators over other potential alternatives. Include a step-by-step explanation of how these operators are applied in the context of their research.

-The properties and lemmas stated require formal proofs to demonstrate their validity.

- There are several issues with the formulas and mathematical notation presented:Some formulas appear to be incomplete or incorrectly formatted (e.g., Eq. 35 and 36). The use of symbols is inconsistent throughout the paper (e.g., π and ϖ are used interchangeably). The authors should provide more context and explanation for each formula presented.

-In equation (1)-(4), how are the parameters π and δ defined and how do they relate to the problem context? What is the recommended range of values for π?

-Equations (7)-(8) define power aggregation operators but do not specify the function used to calculate the similarity measure ω(πj). Provide more details on the choice of similarity function?

-Regarding the bipolar complex fuzzy linguistic operational laws in (10)-(13), could you explain intuitively what each component of the tuples represents (e.g. the positive/negative membership grades)? Some illustrative examples may help interpretation.

-In the score and accuracy functions (14)-(15), what is the justification for using the specific formulae? Have other definitions been considered/evaluated?

- In Eq. 36, what is the significance of the parameter π in the exponents?

-How does the BCFLAAPOWG operator differ from the BCFLAAPOG operator in terms of its mathematical properties and practical applications?

-Can the authors provide a numerical example to illustrate the application of Eq. 35 and 36?

-There are numerous formatting issues, particularly with equations and tables.

-Figures and tables should be properly labeled and referenced in the text.

- No empirical data or case study is presented to demonstrate the practical application of the proposed approach. Examples are needed to validate the methodology works as designed.

- The authors should provide a more comprehensive analysis of their findings. Comparisons with existing methods or techniques should be included to demonstrate the advantages of their proposed approach. The practical implications and limitations of the study should be discussed in detail.

- The conclusion is weak and does not effectively summarize the key contributions of the study: The authors should clearly state the main findings and their significance. Future research directions should be outlined more specifically.

- The paper contains numerous grammatical errors and awkward phrasings that need to be addressed to improve readability.

6. PLOS authors have the option to publish the peer review history of their article (what does this mean?). If published, this will include your full peer review and any attached files.

Reviewer #1: No

Reviewer #2: No

---

## [Author Response · Author response to Decision Letter 0]

31 Jul 2024

Response to the Reviewers/Editor in Chief/Associate Editor in Chief of

PLOS ONE

Subject: PLOS ONE Decision: Revision required 

[PONE-D-24-24446] - [EMID:0c89c56569d8837a]

“Analysis of Coupling in Geographic Information Systems Based on WASPAS Method for Bipolar Complex Fuzzy Linguistic Aczel-Alsina Power Aggregation Operators”

Dear Editors and Reviewers:

First, the authors would like to thank the Editor in Chief, Associate Editor, and anonymous referees for spending their time on the manuscript carefully. The comments of the editors and reviewers are valuable. We have taken all the suggestions/comments positively and did our best to incorporate all these suggestions in the revised version. Our pointwise responses to the reviewer’s comments/suggestions are given below.

Reviewer 1 Comments:

Reviewer #1: 

I have read the manuscript very carefully, the manuscript is very interesting and results

are corrected. After accepting the paper, I have some minor revision, such as:

Response to Reviewer 1# 

Dear Sir/Mam, we appreciate your time in handling our paper and providing suggestions for improvement. We believe the quality of the revised version has considerably improved and hope that you find the revised manuscript satisfactory this time.

Comment 1: Avoid abbreviations in abstract.

Response: Dear Sir/Mam, thank you very much for taking an interest and pointing out these deficiencies. Dear Sir/Mam, we have avoided abbreviations from the abstract section as per your suggestion and highlighted them in the revised manuscript, I hope this time you will be satisfied.

Comment 2: In the introduction section, add advantages, clearly present research gap.

Response: Dear Sir/Mam, thank you very much for pointing out these deficiencies. Dear Sir/Mam, we have revised the introduction section by including the advantages and research gap as per your suggestion and highlighted in the revised manuscript, I hope this time you will be satisfied.

Comment 3: Extended the literature review section by presenting relevant methods, e.g. (parsimonious spherical fuzzy AHP and the integrated IMF SWARA and Fuzzy Bonferroni operator).

Response: Dear Sir/Mam, thank you very much for pointing out these deficiencies. Dear Sir/Mam, we have extended the literature review section (see Ref. [59-62]) as per your suggestion and highlighted in the revised manuscript, I hope this time you will be satisfied.

Comment 4: Present results in figures.

Response: Dear Sir/Mam, thank you very much for pointing out these deficiencies. Dear Sir/Mam, we have done the needful, see Figure 1, I hope this time you will be satisfied.

Comment 5: In Section 2, please include some descriptive analysis for each definition.

Response: Dear Sir/Mam, thank you very much for pointing out these deficiencies. Dear Sir/Mam, we have improved section 2 by including the descriptive analysis for each definition as per your suggestion and highlighted in the revised manuscript, I hope this time you will be satisfied.

Comment 6: The authors suggested checking the grammar carefully.

Response: Dear Sir/Mam, thank you very much for pointing out these deficiencies. Dear Sir/Mam, we have improved the quality of the manuscript especially grammatical mistakes and typos error as per your suggestion and highlighted in the revised manuscript, I hope this time you will be satisfied. 

Comment 7: The order of the citation in your manuscript is not the proper way, please correct it.

Response: Dear Sir/Mam, thank you very much for pointing out these deficiencies. Dear Sir/Mam, we have very carefully revised the order of the citation as per your suggestion and highlighted it in the revised manuscript, I hope this time you will be satisfied.

Comment 8: In the application, add related geometrical diagrams.

Response: Dear Sir/Mam, thank you very much for pointing out these deficiencies. Dear Sir/Mam, we have added the geometrical diagram (see Figure 2) in the application section as per your suggestion and highlighted in the revised manuscript, I hope this time you will be satisfied.

Reviewer 2 Comments:

Reviewer #2: This manuscript presents an approach applying Aczel-Alsina aggregation operators to bipolar complex fuzzy linguistic sets for analyzing coupling in geographic information systems.

However, the paper requires major revisions before it is suitable for publication. I have the following concerns:

Response to Reviewer 2# 

Dear Sir/Mam, we appreciate your time in handling our paper and providing suggestions for improvement. We believe the quality of the revised version has considerably improved and hope that you find the revised manuscript satisfactory this time.

Comment 1: The abstract does not clearly summarize the key contributions and objectives of the paper.

Response: Dear Sir/Mam, thank you very much for taking an interest and pointing out these deficiencies. Dear Sir/Mam, we have very carefully revised the abstract section by including the novelty, motivation, objectives, and major contribution of the proposed work in the abstract section as per your suggestion and highlighted in the revised manuscript, I hope this time you will be satisfied.

Comment 2: The integration of Aczel-Alsina operators with bipolar complex fuzzy linguistic sets appears novel and could provide valuable tools for decision making under uncertainty. However, the novelty claim is not fully justified. - a more extensive literature review is needed to clearly situate this work relative to previous studies applying fuzzy approaches to GIS coupling analysis.

Response: Dear Sir/Mam, thank you very much for pointing out these deficiencies. Dear Sir/Mam, we have very carefully improved the introduction section by including the novelty, motivation, research gap, and problem description, and also improved the literature review section by including some information based on fuzzy approaches to GIS coupling analysis as per your suggestion and highlighted in the revised manuscript, I hope this time you will be satisfied. 

Comment 3: The literature review appears to be incomplete and lacks critical analysis of existing work in the field. The authors should: Expand the review to include more recent publications (within the last 3-5 years). Provide a more in-depth analysis of how their work relates to and builds upon previous research. Clearly identify the research gap their study aims to fill. Add the recent work such as MADM models[10.3389/frai.2024.1347626, 10.22105/jfea.2023.426042.1331, 10.61356/j.mawa.2024.4241, 10.56578/josa020203, 10.1007/s00500-023-09459-0, 10.22105/jarie.2022.339957.1467, 10.31181/sa2120246, 10.22105/jfea.2023.425520.1326, 10.61356/j.mawa.2024.26761]; Bipolar/ linguistic sets[10.1007/s40314-023-02254-5, ]; aggregation operators [10.56578/josa020103, 10.22105/jfea.2023.422582.1318, 10.1007/s40314-021-01651-y, 10.3934/math.2023577, 10.1038/s41598-023-37497-z, 10.22105/bdcv.2021.142090 ]; various fuzzy set extensions[10.1111/exsy.12542, 10.22105/jfea.2024.424633.1328, 10.3934/math.2022954, 10.22105/bdcv.2023.192676, 10.3233/JIFS-210655, 10.22111/ijfs.2024.45118.7968].

Response: Dear Sir/Mam, thank you very much for pointing out these deficiencies. Dear Sir/Mam, we have very carefully revised the literature review section (see Ref. [37-62]) as per your suggestion and highlighted in the revised manuscript, I hope this time you will be satisfied.

Comment 4: The methodology could be explained more clearly, especially the definitions of the various operators which rely on multiple mathematical terms defined earlier. More intuitive explanations may help non-expert readers. Provide a more detailed explanation of the BCFLAAPOA and BCFLAAPOG operators. Justify the choice of these specific operators over other potential alternatives. Include a step-by-step explanation of how these operators are applied in the context of their research.

Response: Dear Sir/Mam, thank you very much for pointing out these deficiencies. Dear Sir/Mam, we have explained the methodology of the proposed work more clearly, especially the definition of the various operators and their properties. We have provided more explanations for the BCFLAAPOA and BCFLAAPOG operators. We have also justified the proposed operators with the help of potential alternatives explained them step-by-step as per your suggestion and highlighted them in the revised manuscript, I hope this time you will be satisfied. 

Comment 5: The properties and lemmas stated require formal proofs to demonstrate their validity.

Response: Dear Sir/Mam, thank you very much for pointing out these deficiencies. Dear Sir/Mam, we have included the proof of the proposed Theorems and Properties as per your suggestion and highlighted in the revised manuscript, I hope this time you will be satisfied.

Comment 6: There are several issues with the formulas and mathematical notation presented: Some formulas appear to be incomplete or incorrectly formatted (e.g., Eq. 35 and 36). The use of symbols is inconsistent throughout the paper (e.g., π and ϖ are used interchangeably). The authors should provide more context and explanation for each formula presented.

Response: Dear Sir/Mam, thank you very much for pointing out these deficiencies. Dear Sir/Mam, we have very carefully revised the formulas and mathematical notions, especially the information in Eq. (35), Eq. (36), and their related problems. Further, we have also explained all the symbols that are used in the proposed manuscript as per your suggestion and highlighted them in the revised manuscript, I hope this time you will be satisfied.

Comment 7: In equation (1)-(4), how are the parameters π and δ defined and how do they relate to the problem context? What is the recommended range of values for π?

Response: Dear Sir/Mam, thank you very much for pointing out these deficiencies. Dear Sir/Mam, the parameters used in Eq. (1) to Eq. (4) play an important role, because, with the help of these parameters, we can easily derive some existing techniques, for instance, if we used the value of π=0, then we will get the model of the drastic t-norm A^d (μ_1,μ_2 ) and drastic t-conorm 〖A^@〗^d (μ_1,μ_2 ), such as

A^d (μ_1,μ_2 )={■(μ_1&if μ_2=1@μ_2&if μ_1=1@0&otherwise)┤

〖A^@〗^d (μ_1,μ_2 )={■(μ_1&if μ_2=0@μ_2&if μ_1=0@1&otherwise)┤

if we used the value of π=π=∞, then we will get the model of the maximum t-norm and minimum t-norm, such as max⁡(μ_1,μ_2 ) and min⁡(μ_1,μ_2 ). But if we use the value of π∈(0,∞), then we get the idea of the Aczel-Alsina t-norm e^(-((-log(μ_1 ))^π+(-log(μ_2 ))^π )^(1/π) ) and Aczel-Alsina t-conorm 1-e^(-((-log(1-μ_1 ))^π+(-log(1-μ_2 ))^π )^(1/π) ), which is the modified version of the algebraic t-norm and t-conorm Where A^π (μ_1,μ_2 )=μ_1*μ_2 and 〖A^@〗^π (μ_1,μ_2 )=μ_1+μ_2-μ_1*μ_2, where the value of δ≥1, I hope this time you will be satisfied.

Comment 8: Equations (7)-(8) define power aggregation operators but do not specify the function used to calculate the similarity measure ω(πj). Provide more details on the choice of similarity function?

Response: Dear Sir/Mam, thank you very much for pointing out these deficiencies. Dear Sir/Mam, we have defined the power aggregation operators in Eq. (7) and Eq. (8), but these definitions are defined for fuzzy values, anyhow we have defined the value of distance measures in Def. (2) and also defined it for BCFL values in Def. (5) as per your suggestion and highlighted in the revised manuscript, I hope this time you will be satisfied.

Comment 9: Regarding the bipolar complex fuzzy linguistic operational laws in (10)-(13), could you explain intuitively what each component of the tuples represents (e.g. the positive/negative membership grades)? Some illustrative examples may help interpretation.

Response: Dear Sir/Mam, thank you very much for pointing out these deficiencies. Dear Sir/Mam, we have explained the information in Eq. (10) to Eq. (13) with the help of some satiable examples as per your suggestion and highlighted in the revised manuscript, I hope this time you will be satisfied.

Comment 10: In the score and accuracy functions (14)-(15), what is the justification for using the specific formulae? Have other definitions been considered/evaluated?

Response: Dear Sir/Mam, thank you very much for pointing out these deficiencies. Dear Sir/Mam, we have easily evaluated the order between any two real numbers, but in the case of complex numbers or in the case of bipolar complex fuzzy linguistic numbers, it is quite complex to describe which one is greater and which is weaker. For evaluating the order between any two BCFL values, we have defined the idea of score values, the concept of score value can easily convert the BCFL number to a simple real number, which can help to order them. But if we obtain the score value of two BCFL numbers that are equal, then we will be using the accuracy values, I hope this time you will be satisfied.

Comment 11: In Eq. 36, what is the significance of the parameter π in the exponents?

Response: Dear Sir/Mam, thank you very much for pointing out these deficiencies. Dear Sir/Mam, we have briefly described the significance of the parameter π in the form of Table 7 before the comparison section and highlighted in the revised manuscript, I hope this time you will be satisfied.

Comment 12: How does the BCFLAAPOWG operator differ from the BCFLAAPOG operator in terms of its mathematical properties and practical applications?

Response: Dear Sir/Mam, thank you very much for pointing out these deficiencies. Dear Sir/Mam, the technique of the BCFLAAPOWG operator and BCFLAAPOG operator are computed based on power aggregation operators, for the construction of the BCFLAAPOG operator, we have used the power operators without weight vector, such as ((1+ω(ϖ_j )))/(∑_(j=1)^n▒(1+ω(ϖ_j )) ), but in the construction of the BCFLAAPOWG operators, we have used the power operators with weight vectors, such as (Ψ_j (1+ω(ϖ_j )))/(∑_(j=1)^n▒〖Ψ_j (1+ω(ϖ_j )) 〗), which can affect the final results, see the ranking values in Table 6 and Table 8, it means that the weight vector plays an important role in the construction of the above operators, but in mathematical properties, both operators are the same, I hope this time you will be satisfied.

Comment 13: Can the authors provide a numerical example to illustrate the application of Eq. 35 and 36?

Response: Dear Sir/Mam, thank you very much for pointing out these deficiencies. Dear Sir/Mam, we have illustrated numerical examples for the aggregation operators in Eq. (35) and Eq. (36) as per your suggestion and highlighted in the revised manuscript, I hope this time you will be satisfied.

Comment 14: There are numerous formatting issues, particularly with equations and tables.

Response: Dear Sir/Mam, thank you very much for pointing out these deficiencies. Dear Sir/Mam, we have very carefully revised the formatting problems of the equations, Tables, and Figures as per your suggestion and highlighted in the revised manuscript, I hope this time you will be satisfied.

Comment 15: Figures and tables should be properly labeled and referenced in the text.

Response: Dear Sir/Mam, thank you very much for pointing out these deficiencies. Dear Sir/Mam, we have properly labeled all the Figures and Tables in the revised manuscript, I hope this time you will be satisfied.

Comment 16: No empirical data or case study is presented to demonstrate the practical application of the proposed approach. Examples are needed to validate the methodology works as designed.

Response: Dear Sir/Mam, thank you very much for pointing out these deficiencies. Dear Sir/Mam, we have briefly discussed the case study and also demonstrated the practical application of the proposed work. Further, we have also given some examples for showing the validation of the proposed methodology as per your suggestion and highlighted in the revised manuscript, I hope this time you will be satisfied.

Comment 17: The authors should provide a more comprehensive analysis of their findings. Comparisons with existing methods or techniques should be included to demonstrate the advantages of their proposed approach. The practical implications and limitations of the study should be discussed in detail.

Response: Dear Sir/Mam, thank you very much for pointing out these deficiencies. Dear

---

## [Decision Letter · Decision Letter 1]

21 Aug 2024

Analysis of Coupling in Geographic Information Systems Based on WASPAS Method for Bipolar Complex Fuzzy Linguistic Aczel-Alsina Power Aggregation Operators

PONE-D-24-24446R1

Dear Dr. Pamucar,

We’re pleased to inform you that your manuscript has been judged scientifically suitable for publication and will be formally accepted for publication once it meets all outstanding technical requirements.

Kind regards,

Muhammet Gul, Ph.D.

Academic Editor

PLOS ONE

Additional Editor Comments (optional):

Reviewers' comments:

Reviewer's Responses to Questions

**Comments to the Author**

1. If the authors have adequately addressed your comments raised in a previous round of review and you feel that this manuscript is now acceptable for publication, you may indicate that here to bypass the “Comments to the Author” section, enter your conflict of interest statement in the “Confidential to Editor” section, and submit your "Accept" recommendation.

Reviewer #1: (No Response)

Reviewer #2: All comments have been addressed

2. Is the manuscript technically sound, and do the data support the conclusions?

Reviewer #1: (No Response)

Reviewer #2: Yes

3. Has the statistical analysis been performed appropriately and rigorously? 

Reviewer #1: (No Response)

Reviewer #2: N/A

4. Have the authors made all data underlying the findings in their manuscript fully available?

Reviewer #1: (No Response)

Reviewer #2: Yes

5. Is the manuscript presented in an intelligible fashion and written in standard English?

Reviewer #1: (No Response)

Reviewer #2: Yes

6. Review Comments to the Author

Reviewer #1: (No Response)

Reviewer #2: The authors have adequately addressed the concerns and suggestions raised in the previous review. The revised manuscript reflects significant improvements and meets the necessary standards for publication. Therefore, I recommend the acceptance of the revised version for publication.

7. PLOS authors have the option to publish the peer review history of their article (what does this mean?). If published, this will include your full peer review and any attached files.

Reviewer #1: No

Reviewer #2: No

---

## [Editor Report · Acceptance letter]

23 Aug 2024

PONE-D-24-24446R1 

PLOS ONE

Dear Dr. Pamucar, 

I'm pleased to inform you that your manuscript has been deemed suitable for publication in PLOS ONE. Congratulations! Your manuscript is now being handed over to our production team.

Kind regards, 

on behalf of

Dr. Muhammet Gul 

Academic Editor

PLOS ONE